# Accurate space-based NOx emission estimates with the flux divergence approach require fine-scale model information on local oxidation chemistry and profile shapes

Felipe Cifuentes[1,2], Henk Eskes[1], Enrico Dammers[3,4], Charlotte Bryan[1], and Folkert Boersma[2,1]

[1]R&D Satellite Observation department, Royal Netherlands Meteorological Institute (KNMI), De Bilt, 3731 GA, The Netherlands
[2]Meteorology and Air Quality department, Wageningen University & Research (WUR), Wageningen, 6708 PB, The Netherlands
[3]Air Quality and Emissions Research, Netherlands Organisation for Applied Scientific Research (TNO), Utrecht, 3584 CB, The Netherlands
[4]Institute of Environmental Sciences (CML), Leiden University, Leiden, 2333 CC, The Netherlands

**Correspondence:** Felipe Cifuentes (felipe.cifuentescastano@knmi.nl)

**Abstract.** The flux divergence approach (FDA) is a popular technique for deriving $NO_x$ emission estimates from tropospheric $NO_2$ columns measured by the TROPOMI satellite sensor. An attractive aspect of the FDA is that the method simplifies three-dimensional atmospheric chemistry and transport processes into a two-dimensional (longitude-latitude) steady-state continuity equation for columns that balances local $NO_x$ emissions with the net outflow and chemical loss of $NO_x$. Here we test the capability of the FDA to reproduce known $NO_x$ emissions from synthetic $NO_2$ column retrievals generated with the LOTOS-EUROS chemistry transport model over the Netherlands at high spatial resolution of about 2x2 km during summer. Our results show that the FDA captures the magnitude and spatial distribution of the $NO_x$ emissions to high accuracy (absolute bias <9%), provided that the observations represent the $NO_2$ column in the boundary layer, that wind speed and direction are representative for the boundary layer (PBL) column, and that the high resolution spatiotemporal variability of the $NO_2$ lifetimes and $NO_x$:$NO_2$ ratio is accounted for in the inversion, instead of using single fixed values. The FDA systematically overestimates $NO_x$ emissions by 15-60% when using tropospheric $NO_2$ columns as the driving observation, while using PBL $NO_2$ columns largely overcomes this systematic error. This merely reflects that the local balance between emissions and sinks of $NO_x$ occurs in the boundary layer, which is decoupled from the $NO_2$ in the free troposphere. Based on the recommendations from this sensitivity test, we then applied the FDA using observations of $NO_2$ columns from TROPOMI, corrected for contribution from free tropospheric $NO_2$, between 1 June and 31 August 2018. The $NO_x$ emissions derived from the default TROPOMI retrievals are biased low over cities and industrialized areas. However, when the coarse 1x1 degree TM5-MP $NO_2$ profile used in the retrieval is replaced by the high-resolution profile of LOTOS-EUROS, the TROPOMI $NO_x$ emissions are enhanced by 22% and are in better agreement with the inventory for the Netherlands. This emphasizes the importance of using realistic high-resolution *a-priori* $NO_2$ profile shapes in the TROPOMI retrieval. We conclude that accurate quantitative $NO_x$ emissions estimates are possible with the FDA, but that they require sophisticated, fine-scale, corrections for both the $NO_2$ observations driving the method, as well as the estimates of the $NO_2$ chemical lifetime and $NO_x$:$NO_2$ ratio.

This information can be obtained from high-resolution chemistry transport model simulations, at the expense of the simplicity and applicability of the FDA.

*Copyright statement.* TEXT

## 25    1   Introduction

Nitrogen oxides ($NO_x = NO + NO_2$) are highly reactive atmospheric trace gases, primarily originating from fossil-fuel combustion in mobile and industrial sources, as well as biomass burning, microbial activity in soils, and lighting (Song et al., 2021; Murray, 2016). These compounds contribute to the formation of tropospheric ozone and secondary aerosols (Seinfeld and Pandis, 2006); thereby causing negative implications for human health, climate, and terrestrial and aquatic ecosystems (Clark
et al., 2013; de Vries, 2021). Accurate and regularly updated emission inventories combined with observations are needed to assess the current pollution levels, formulate control measures, and track their effectiveness.

Conventional bottom-up approaches to estimate $NO_x$ emissions rely on combining aggregated activity data, average emission factors, and spatiotemporal proxies for disaggregation. These methods have considerable uncertainties due to factors such as omitted sources, an incomplete comprehension of sectoral activity, real-world operating conditions, and spatial distribution
of sources (Pommier, 2022; Lonsdale and Sun, 2023; Liu et al., 2022). Moreover, the estimates are outdated by at least a year, due to the time required to collect the data (Wang et al., 2020; Zhang et al., 2023).

In contrast, satellite observations, available in real-time, provide comprehensive and independent information about the global distribution of the total amount of $NO_2$ in the atmosphere with city-scale resolution, allowing the quantification of major point sources (Beirle et al., 2021, 2023; Chen et al., 2023; Dammers et al., 2022; Fioletov et al., 2022). These observations
can be linked to emissions by accounting for the chemical conversion and transport of the atmospheric $NO_2$ (Liu et al., 2022). The satellite-based emissions hence provide up-to-date information obtained by observing real operating conditions, can track the time dependence of emissions, and facilitate the identification of overlooked sources due to their full spatial coverage (Pommier, 2022; Lorente et al., 2019).

Despite providing valuable insights, satellite-derived emissions have limitations. $NO_2$ observations are limited to clear-sky
conditions and only represent the atmospheric conditions close to the satellite overpass time. For polar sun-synchronous orbit satellites, the overpass occurs at a fixed local time, which excludes the estimation of diurnal emission profiles and may overlook significant sources inactive during the observations. Additionally, current satellites maintain a relatively coarse spatial resolution compared to specific regional bottom-up emission inventories that achieve kilometer or sub-kilometer scale. Satellite-derived emissions therefore supplement traditional bottom-up emission inventories rather than replace them, offering
additional and complementary information, and a benchmark for validation because they are fully independent.

The TROPOspheric Monitoring Instrument (TROPOMI) (Veefkind et al., 2012) has been widely employed for deriving satellite-based $NO_x$ emissions. The revolutionary pixel size at nadir of 3,5 km x 7 km (improved to 3,5 km x 5,5 km after

August 2019), and the high signal-to-noise ratio, make the product suitable for examining emissions from diverse sources such as city emissions (Lorente et al., 2019; Pommier, 2022; Xue et al., 2022; Zhang et al., 2023), power plants (Goldberg et al., 2019; Saw et al., 2021; Skoulidou et al., 2021a; Krol et al., 2024), oil and gas production (Dix et al., 2022), individual ships (Georgoulias et al., 2020; Kurchaba et al., 2022; Riess et al., 2024), lighting (Allen et al., 2021; Zhang et al., 2022), soil (Lin et al., 2023) and croplands (Huber et al., 2020). Beyond the $NO_x$ emissions, the TROPOMI instrument has also been employed to derive emission datasets for $CH_4$ (Liu et al., 2021), $SO_2$ (Chen et al., 2024; Fioletov et al., 2020) and CO (Leguijt et al., 2023).

Diverse methods have been employed to estimate satellite-based emissions. Chemical transport models (CTM) based approaches use satellite data as a constraint to enhance the emission inventory, using techniques such as mass balance (Cooper et al., 2017), variational data assimilation (Yarce Botero et al., 2021), Kalman filters (Ding et al., 2017), and analytical inversion (Lu et al., 2022). While CTM-based methods incorporate detailed three-dimensional chemical and meteorological processes allowing to obtain spatiotemporally resolved emissions, their application is constrained by the requirement for additional input datasets and computational expenses (Lonsdale and Sun, 2023). On the other hand, CTM-independent approaches are based on plume dispersion models or local mass conservation applying a steady-state continuity equation. Plume dispersion methods involve fitting a dispersion model to the concentration data across the affected area to derive emissions, requiring a precise definition of the region influenced by plume advection. In contrast, local mass conservation methods operate at the source or pixel level, which eliminates the need to define a specific region of interest and allows for the simultaneous estimation of multiple sources (Misra et al., 2021).

Beirle et al. (2019) proposed a flux divergence approach (FDA) based on the steady-state continuity equation for $NO_2$ columns to extract surface $NO_x$ emissions. This method uses satellite observations of $NO_2$ vertical column densities (VCD) and requires knowledge of horizontal wind components, a conversion factor from $NO_2$ to $NO_x$, and the chemical lifetime of $NO_2$ at satellite overpass time. Different authors applied this methodology to estimate $NO_x$ emissions at the global scale (Beirle et al., 2023) or for specific regions, such as the United States (Dix et al., 2022), North India (Misra et al., 2021), Egypt (Rey-Pommier et al., 2022), South Asia (de Foy et al., 2022), and Taiwan (Chen et al., 2023). In addition, few studies have evaluated the accuracy and limitations of FDA satellite-derived emissions by using synthetic observations generated through chemical transport models as input within their approach (Dix et al., 2022; Hakkarainen et al., 2022; Goldberg et al., 2022). In contrast, most other studies have primarily focused on evaluating the FDA's sensitivity to different parameters and on estimating uncertainties arising from input data. The major sources of uncertainty in the method are considered to be biases in the observed $NO_2$ VCD, due to reduced vertical sensitivity to near-surface $NO_2$ by the satellite products, and the influence of selected *a-priori* $NO_2$ vertical profile shapes (Douros et al., 2023). Additional sources of uncertainty include systematic biases in the zonal and meridional wind components and the altitude at which they are sampled, and faulty representation of the spatial and seasonal changes in $NO_x$ lifetime and the $NO_2$ to $NO_x$ conversion factors (Beirle et al., 2019, 2021, 2023).

This study aims to evaluate the accuracy with which known $NO_x$ emissions can be reproduced by the FDA, identify its weaknesses and uncertain parameters, and provide recommendations for its application and improvement. Synthetic $NO_2$ VCD satellite observations at high spatial resolution, generated with the LOTOS-EUROS model (Schaap et al., 2008; Manders

et al., 2017), were used to derive a $NO_x$ emission dataset using the FDA. The new dataset was compared qualitatively and quantitatively to the emissions originally ingested into the model to characterize the performance and quantify the various contributions to the uncertainty of the method. This study also delves into the impact of using high spatial resolution (2x2 km$^2$) chemical and meteorological fields to represent the variability of the $NO_2$ VCD profile shapes, $NO_2$ lifetime, and the $NO_2$ to $NO_x$ conversion factors, and studies their importance in reconstructing accurate $NO_x$ emission datasets. Furthermore, we will evaluate the applicability of the FDA for satellite observations at different times of day, different from the noon conditions of the TROPOMI overpass time, to assess the method's applicability throughout the day. This provided insights into the suitability of this method for geostationary observations.

## 2   Data

### 2.1   TROPOMI $NO_2$ tropospheric column

The TROPOMI instrument, on board of the Sentinel-5 Precursor (S-5 P) polar satellite, is a nadir-viewing spectrometer. It measures radiation across the ultraviolet, visible, and infrared spectral regions and is utilized for monitoring atmospheric trace gases and aerosols (Veefkind et al., 2012). $NO_2$ columns are retrieved following a three-step procedure. First, the $NO_2$ slant column density is derived from the L1b spectra measured by TROPOMI using a Differential Optical Absorption Spectroscopy (DOAS) fit. The slant column is divided into a stratospheric and tropospheric fraction using data assimilation within the TM5-MP model at a 1x1°horizontal resolution (Williams et al., 2017). Lastly, the slant columns are converted into VCD using total and altitude-dependent air mass factors (AMFs). The AMFs are dependent on the $NO_2$ vertical profiles derived from TM5-MP, the viewing geometry of the satellite, the surface albedo, surface pressure, and clouds and aerosols characteristics. Further description of the retrieval can be consulted at van Geffen et al. (2022a) and van Geffen et al. (2022b).

Routine validation of TROPOMI $NO_2$ observations against ground-based MAX-DOAS measurements at 29 stations revealed a mean bias of -28%, escalating to -40% over heavily polluted regions (Lambert et al., 2023). This bias is predominantly linked to the vertical profile generated by TM5-MP, which insufficiently resolves concentration hotspots and exhibits deviations in profile shape, particularly near the Earth's surface (Chan et al., 2020; Verhoelst et al., 2021). The discrepancies in TROPOMI observations can then be partially mitigated by updating the *a-priori* vertical profile with one derived from a higher-resolution air quality model (Griffin et al., 2019; Zhao et al., 2020; Judd et al., 2020; Douros et al., 2023). This procedure is done via TROPOMI averaging kernels and it is explained in the TROPOMI $NO_2$ Product User Manual (Eskes et al., 2022). Remaining biases in TROPOMI observations might arise from errors in the slant column, the stratospheric-tropospheric partitioning, and various factors affecting the air mass factor (AMF) beyond the *a-priori* profile, including surface albedo and cloud cover.

This study used the TROPOMI L2 $NO_2$ version 2.4.0 reprocessed product, focusing on orbits over the Netherlands from 1 June to 31 August 2018. During this period, TROPOMI's nadir pixel size was 3,5 km x 7 km. The equator overpass occurs around 13:30 local time. To enhance the data reliability, pixels with a quality assurance value below 0,75 were excluded, effectively removing pixels with cloud radiance fractions higher than 0,5 and minimizing the impact of uncertain retrievals (van Geffen et al., 2022a). Additionally, a local TROPOMI product was generated by replacing the TM5-MP *a-priori* with the

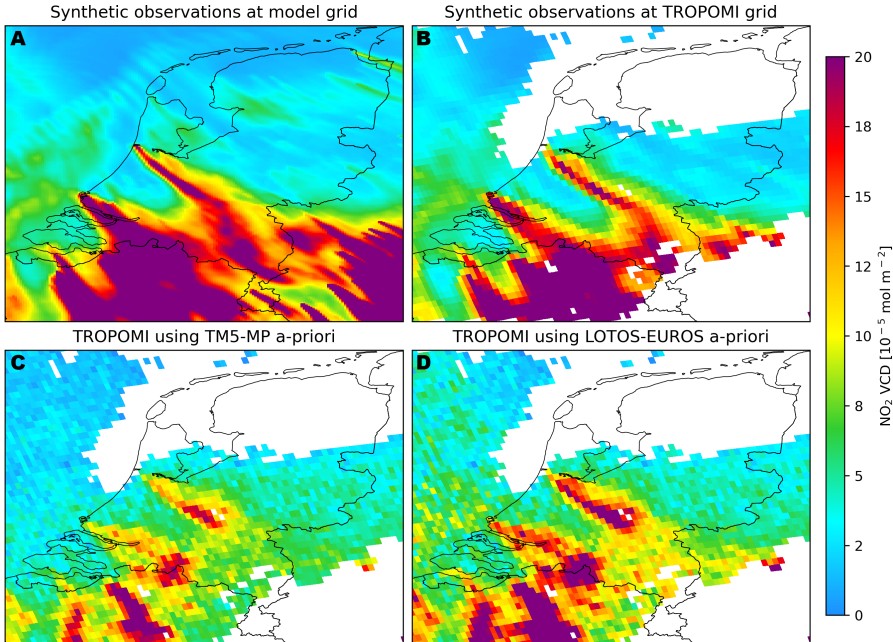

**Figure 1.** Comparison of $NO_2$ vertical column densities over the Netherlands on 5 July 2018, derived from synthetic and actual TROPOMI observations. **(A)** Synthetic observations at native LOTOS-EUROS resolution ($2x2$ $km^2$). **(B)** Synthetic observations spatiotemporally interpolated into the TROPOMI grid, and excluding the pixels where TROPOMI observations do not satisfy a quality assurance value over 0,75. **(C)** TROPOMI observations using the default TM5-MP *a-priori* profile shape. **(D)** TROPOMI observations using a high-resolution *a-priori* profile shape derived from the LOTOS-EUROS simulation.

one derived from the LOTOS-EUROS high-resolution simulations over the Netherlands (See section 2.2). Figure 1 show the variation in $NO_2$ VCD between the TM5-MP and LOTOS-EUROS TROPOMI products, showing a 23% increase in TROPOMI observed $NO_2$ VCD over heavily polluted areas when using LOTOS-EUROS *a-priori* $NO_2$ profiles. In contrast, background regions do not show significant enhancements.

## 2.2 Synthetic $NO_2$ observations

LOTOS-EUROS is an offline 3D CTM developed in the Netherlands. The model is used for operational forecast in the Netherlands and Europe (Manders et al., 2017). Its utility extends to research applications in diverse global regions, such as northwest South America (Yarce Botero et al., 2021) and China (Timmermans et al., 2017; Petersen et al., 2019). LOTOS-EUROS is part of the Copernicus Atmospheric Monitoring Service (CAMS) European air quality ensemble, a service that provides forecasts and reanalyses of the main air pollutants using 11 state-of-the-art CTMs. Within CAMS, LOTOS-EUROS undergoes routine validation with in-situ observations and TROPOMI satellite data, and is evaluated against the other ensemble members (Peuch et al., 2022). Additionally, independent studies have demonstrated good agreement between the simulated tropospheric $NO_2$

columns and those measured by TROPOMI and ground-based remote sensing instruments in the Netherlands (Vlemmix et al., 2015) and over Greece (Skoulidou et al., 2021b). These studies found discrepancies ranging from 1% to 35% when comparing

LOTOS-EUROS columns with measurements from MAX-DOAS instruments. LOTOS-EUROS has also participated in numerous model intercomparison studies showing overall strong performance (Bessagnet et al., 2016; Colette et al., 2017; Vivanco et al., 2018).

In this study, the LOTOS-EUROS v2.2.009 model was used to generate an hourly high-resolution (2x2 $km^2$) $NO_2$ dataset across the Netherlands spanning from 1 June to 31 August 2018. The simulations were conducted using 12 vertical levels,

extending from the ground to approximately 9 km above the earth's surface. Tropopause heights can exceed 15 km in tropical regions and typically between 8 and 12 km in other areas (Xian and Homeyer, 2019). Given that this study focuses on the Netherlands, using vertical layers up to 9 km should provide a reasonable estimate of the tropospheric $NO_2$ column, as the majority of $NO_2$ is contained within this altitude range.

The simulations followed a one-way nesting approach with 3 nested domains. The parent domain covered Europe (15°W-

145 35°E; 35-70°N), the intermediate domain focused on North-western Europe (2-16°E; 47-56°N), and the target domain covered the Netherlands (3.1-7.5°E; 50.3-53.7°N) with a horizontal resolution of 2x2 $km^2$. The model was run using the European Center for Medium-Range Weather Forecast (ECMWF) Integrated Forecast System (IFS) as the meteorological driver. The emissions used at the European scale were taken from the CAMS-REG-v5.1 inventory, whereas the emissions for the Netherlands domain were a combination of the CAMS-REG-v5.1 inventory, with the Dutch and German emissions replaced by the

150 national GrETa and ER emission inventories (in both cases using reported emissions of 2018). The emission inventories previously mentioned consist of annual total estimates, which were distributed using monthly, daily, and hourly time factors for different aggregated source categories. Emissions are also distributed vertically, with specific heights assigned on a sector-by-sector basis. In particular, industrial sources and public power stations have vertical distributions based on typical average stack heights. Further details can be found in Manders et al. (2021).

Two types of synthetic datasets were derived from the modeling outputs. (1) A synthetic dataset at the native CTM resolution and grid, temporally interpolated to 13:30 LT to align with TROPOMI's overpass time, and (2) a synthetic dataset of $NO_2$ VCD spatiotemporally interpolated into the TROPOMI grid and timestamp for each of the orbits that crossed the Netherlands during the period of analysis, and excluding the pixels where and when TROPOMI observations do not satisfy a quality assurance value over 0.75. Figure 1 shows the different types of datasets, demonstrating the variation in resolution and the absence of

valid data in the synthetic dataset at the TROPOMI grid, where clouds and problematic retrievals occurred.

We compared LOTOS-EUROS synthetic $NO_2$ VCD with TROPOMI daily observations, finding good agreement in the shape, direction, and extent of plumes for major hotspots in the Netherlands, North Belgium, and West Germany. Figure A1 illustrates two examples of these comparisons. The high-resolution simulations of our simulations oversample TROPOMI's resolution by a factor of two, providing a more detailed representation of the chemistry within the plumes.

## 2.3 Meteorological and chemical inputs for the FDA

Temperature, planetary boundary layer (PBL) height, and zonal and meridional components of the wind at various vertical levels were extracted from the LOTOS-EUROS meteorological files, which were regridded from ECMWF-IFS operational forecast data. Only the data that was downscaled to the innermost nested domain was used to have high-resolution information ($2x2$ km$^2$). In addition, OH concentrations at various vertical levels and NO VCDs were extracted from the LOTOS-EUROS outputs.

## 3 Methods

### 3.1 Emissions estimation using the flux divergence method

Following the implementation of the steady-state continuity equation proposed by Beirle et al. (2019),

$$E = D + S = \nabla(LV\mathbf{w}) + \frac{LV}{\tau} \tag{1}$$

where the $NO_x$ emissions ($E$) are computed as the sum of the divergence of the $NO_x$ flux ($D$) and a sink term ($S$). The $NO_x$ flux can be expressed as $LV\mathbf{w}$, where $V$ is the tropospheric $NO_2$ VCD observation, $L$ is a conversion factor from $NO_2$ to $NO_x$, and $\mathbf{w}$ represents the wind field. The sink term $S$ can be represented as $LV/\tau$, where $\tau$ is the lifetime of $NO_2$ at overpass time. The lifetime depends on the rate of loss of $NO_x$, which can occur through chemical reactions, deposition, and dispersion (Griffin et al., 2021). However, during daylight, the primary mechanism for $NO_x$ loss is its chemical reaction with hydroxyl radicals (OH), resulting in the formation of nitric acid ($HNO_3$) (Lange et al., 2022).

The divergence term $\nabla(LVw)$ can be estimated on a grid using a fourth-order central-finite difference,

$$\nabla\mathbf{f}(x) = \frac{f(x-2h) - 8f(x-h) + 8f(x+h) - f(x+2h)}{12h}, \tag{2}$$

where *f(x)* represents *LVw* and h is the spacing between observations. Note that other finite difference methods can be used. As part of the preliminary test conducted on this study, a second-order method including the nearest neighbors in the east-west and north-south directions, and a second order including the neighbors in the diagonal direction were conducted. The impact of changing the finite difference scheme was minor, within 0,5% for normalized bias and gross error, and 0,02 in correlation. Koene et al. (2024) recommend calculating the divergence term using the smallest possible stencil to reduce noise impact. Consequently, applying a second-order finite difference method or estimating flux at cell boundaries might be more effective when processing noisy data. However, these improvements were not discernible in this study, as the synthetic $NO_2$ fields used are noise-free. All reported results below are based on the fourth-order approach, to keep consistency with the original implementation of the FDA. Further details on the assumptions or methods used to select all the variables discussed previously are presented below.

### 3.1.1 Steady-state assumption

The steady-state assumptions imply the absence of accumulation or depletion of atmospheric $NO_x$ concentrations within the analyzed area. Factors such as turbulent mixing, changes in wind patterns, variations in emission sources, and sinks can disturb this balance and create fluctuations of $NO_x$ within the plumes (Koene et al., 2024). However, when averaged over time and across different realizations of the turbulence, the influence of these variations can be reduced. For instruments like TROPOMI, stability in typical overpass conditions is assumed. Indeed, Li et al. (2021) examined the daily variation of $NO_2$ and demonstrated that during the period from 12:00 to 14:00, $NO_x$ emissions and mixing layer heights exhibit comparable levels, resulting in stable $NO_2$ concentrations, which support the assumption of a steady state during/around the overpass of TROPOMI.

### 3.1.2 Wind fields

The application of the FDA requires the reduction of the three-dimensional transport of pollutants in the atmosphere into a two-dimensional space, involving the estimation of effective zonal and meridional wind fields (or along and across track when using satellite grids directly). These effective wind components should be profile-weighted, which requires prior knowledge of the $NO_2$ and wind profiles within the column (Koene et al., 2024). As an approximation, Lorente et al. (2019) used $NO_2$ weighted and unweighted mean boundary layer wind fields, while Bryan (2022) proposed to dynamically extract the wind components based on a specific fraction of the PBL (half PBL height), which works well under the assumption of well-mixed $NO_2$ within the PBL. Alternatively, some authors proposed using a fixed altitude below the PBL height at the time of the satellite overpass, such as 450 m (Beirle et al., 2019), 300 m (Beirle et al., 2021), 100 m (de Foy et al., 2022; Goldberg et al., 2022) and 80 m (Misra et al., 2021). Nonetheless, this approach neglects the day-to-day variability and spatial patterns of the PBL, affecting the $NO_2$ and winds vertical distribution within the columns.

The wind divergence becomes non-zero when more air leaves a vertical column than enters it, and it is a phenomenon induced by global-scale processes (such as vertical transport and transport between high and low-pressure areas), large-scale features (like mountains and coastlines), and due to numerical interpolation (Bryan, 2022). Moreover, simplifying the three-dimensional structure into a two-dimensional representation on the FDA leads to a violation of the conservation of air mass. It is therefore important to address wind divergence effects. Removing the divergence from the 2D wind field ensures that the air mass of the 2D total column field is locally conserved. Bryan (2022) outlined an iterative algorithm to generate a wind dataset with reduced divergence by making slight adjustments to the wind fields. The method is similar to the Newton-Rhapson technique which iteratively approximates the minimum of a function by descending along the gradient of the function. For a comprehensive description of the method, readers are directed to the study by Bryan (2022)

### 3.1.3 NO$_2$ lifetime ($\tau$)

During satellite overpass time (13:30 LT for TROPOMI), the main mechanism for NO$_x$ chemical loss is the reaction of NO$_2$ with OH to form HNO$_3$.

$$\text{OH} + \text{NO}_2 + \text{M} \rightarrow \text{HNO}_3 + \text{M} \tag{R1}$$

This reaction is characterized by a lifetime $\tau$, and in its original implementation, Beirle et al. (2019) used a constant $\tau$ of 4 hours to estimate NO$_x$ emissions in Riyadh, South Africa, and Germany. This value was derived from the analysis of the downwind plume from Riyadh and generalized as an average representative value of the NO$_2$ decay from megacities and power plants as observed from satellite instruments. However, this approach neglects the nonlinear dependency between $\tau$ and NO$_x$
concentrations (Laughner and Cohen, 2019; Valin et al., 2013), as well as the dependency on the photolysis rate, temperature, and relative humidity (Beirle et al., 2003; Misra et al., 2021). To address this complexity, a recent study (Rey-Pommier et al., 2022) has employed a first-order kinetic equation to represent the rate constant for the reaction R1,

$$\tau = \frac{1}{K[OH]} = \frac{1}{2.8e^{-11}\left(\frac{T}{300}\right)^{-1.3}[OH]}, \tag{3}$$

where $\tau$ is computed from temperature ($T$ [K]) and the OH concentration (OH in molecules cm$^{-3}$]). In Rey-Pommier et al.
(2022) OH was extracted from global model simulations by CAMS. Lifetime estimates using this method are influenced by the resolution of the CTM, owing to the nonlinear production and loss of NO$_x$. A CTM running at fine resolution can result in extended lifetimes in NO$_x$-saturated regions due to enhanced OH titration by NO$_x$. Conversely, in NO$_x$-limited regimes, it can result in shorter lifetimes as elevated VOC levels promote OH production in the presence of the available NO$_X$ (Li et al., 2023; Krol et al., 2024).

### 3.1.4 NO$_x$ deposition

Another sink for NO$_x$ is its removal from the atmosphere through dry and wet scavenging processes. Since the application of the FDA to satellite images is limited to clear sky conditions, only dry deposition, where NO$_x$ is directly transferred from the atmosphere to surfaces such as soil and vegetation, impacts the estimation of emissions using the FDA. However, (Rey-Pommier et al., 2022) indicated that the lifetimes associated with deposition are about an order of magnitude larger than the
245 chemical lifetimes, making the deposition contributions to the sink less significant, as the sink is proportional to the inverse of the lifetime. Furthermore, only surface NO$_x$ is subjected to deposition, whereas the entire column is exposed to chemical loss. In our study, we did not account for the effect of deposition on the emission inversions using the FDA. Note that this condition applies at noon when photochemistry is enhanced due to higher incoming solar radiation. At other times of the day, the contribution of deposition sinks relative to chemical sinks can become more significant.

### 3.1.5 NOx partitioning factor

Similarly to $\tau$, the partition between NO and $NO_2$ in the atmosphere is influenced by factors such as the actinic flux, ozone concentrations, and temperature. A constant value of $L = 1,32$ is often assumed (Beirle et al., 2019; de Foy et al., 2022; Misra et al., 2021), which is considered representative of the usual satellite observation conditions (noon time and cloud-free pixels). However, as noted by Hakkarainen et al. (2024), recent studies increasingly use model-derived values for $L$. This includes simulations from global models like CAMS (Lorente et al., 2019; Rey-Pommier et al., 2022) and regional models (Goldberg et al., 2022). Alternatively, Beirle et al. (2021) derived $L$ values using a photo-stationary steady-state approach. These methods have reported $L$ values ranging from 1,16 to 1,83, deviating from the typical mean of 1,32. Large eddy simulations (LES) further indicate that $L$ can rise as high as 5 within the first 10 km of emitted plumes (Krol et al., 2024).

## 3.2 FDA test using synthetic observations

Quantifying the accuracy of FDA-derived emissions is challenging as the true emission values are unknown. An alternative approach involves conducting an end-to-end test (E2E), schematically represented in Fig 2. This entails employing synthetic satellite observations generated within a chemical transport model (LOTOS-EUROS) to assess the FDA capability to reconstruct the emissions used as input, thereby evaluating the performance of the method. Additionally, by varying the input data and configurations of the FDA it is possible to assess the sensitivity of the method and identify the setting that enhances the accuracy of the derived emissions. The specific tests conducted in this study are described below.

### 3.2.1 Overall performance assessment and sensitivity analyses at CTM resolution

The FDA was applied to the $NO_2$ VCD synthetic observations at the native CTM resolution (2x2 $km^2$) to estimate $NO_x$ emissions on a daily basis, spanning from 1 June to 31 August 2018, and subsequently averaged into the three-monthly mean for June-July-August (JJA). The $NO_2$ VCD at LOTOS-EUROS native resolution allows testing the FDA in ideal conditions. No quality screening is applied, ensuring full coverage of $NO_2$ VCD synthetic observations (See Figure 1A). Observations are available on the model grid, avoiding interpolations in the observation operator, and the kernels are idealized as unit vector (1,1,..,1) so the observations correspond to true total columns.

Seven tests, summarized in Table 1, were conducted to assess the FDA's overall performance and examine its sensitivity to different input data and configurations. Tests 01 and 02 were conducted to evaluate the impact of using the full LOTOS-EUROS modeled column versus only the column within the PBL as a more accurate proxy for representing instantaneous emissions at satellite overpass time. Under typical noon conditions, which correspond to the TROPOMI overpass time, emissions are generally contained within the PBL. Therefore we consider using PBL-integrated columns to be a valid approach. However, in the early morning, when the PBL is still forming, some emissions may extend beyond the PBL, a phenomenon that is prone to occurring only during winter periods. To the best of our knowledge, applications of the FDA approach in the peer-reviewed literature have exclusively used $NO_2$ tropospheric columns without critically considering that free tropospheric $NO_2$ is decoupled from surface emissions. Liu et al. (2021) excluded the free tropospheric contribution in their analysis of methane

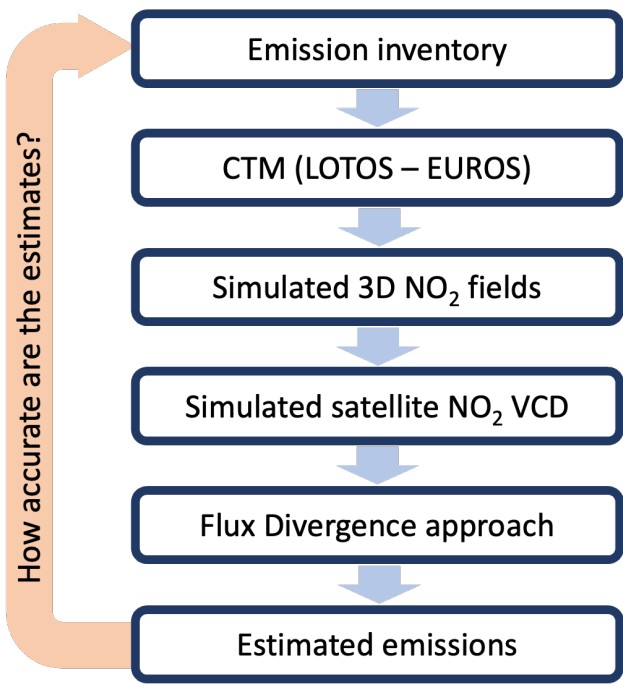

**Figure 2.** Schematic representation of the evaluation system designed to assess the accuracy of the $NO_x$ emissions derived using the flux divergence approach

emissions, while Hakkarainen et al. (2022) applied a similar approach for $CO_2$. Tests 02-05 assessed the impact of using different fixed-altitude wind fields compared to dynamically selected wind fields based on PBL height. After identifying the optimal approach for selecting wind fields, Test ID06 evaluated the impact of using dynamic lifetimes and $NO_2$ to $NO_x$

conversion values derived from the LOTOS-EUROS simulation as opposed to fixed-single values. Finally, Test ID07 assessed the impact of wind fields corrected for divergence on the final emission estimates.

The accuracy in each scenario was evaluated qualitatively, through comparisons between the JJA emissions map for the original and derived emissions at 13:30 LT, and quantitatively, computing statistical metrics such as correlation coefficient (R), normalized mean bias (NMB), and normalized mean gross error (NMGE). These metrics were calculated at the pixel level for

the entire simulated domain and for the hotspot subset separately, defined as pixels containing the top 10% percentile emission values of the original emission inventory. Detailed definitions of these statistical metrics can be found in Appendix A.

A second set of performance metrics was estimated by comparing the FDA-derived emissions with a convoluted version of the model-ingested emissions, using a 3x3 grid cell spread function of the form ([1/16, 1/8, 1/16], [1/8, 1/4, 1/8], [1/16, 1/8, 1/16]). This convolution spreads the original emissions into neighboring cells, mimicking the numerical smearing caused by

the numerical solution of the divergence term in the FDA, enabling a more fair comparison between datasets. With this kernel a point emission is effectively distributed over 4 grid cells. Figure 3 shows the original and convoluted model-ingested emissions alongside the FDA-derived emissions in Test ID06, illustrating how the datasets compare in each scenario. When compared to

**Table 1.** Summary of sensitivity tests applied using the $NO_2$ VCD at native model resolution.

| ID | Column integration height | Winds altitude | Lifetime | $NO_x/NO_2$ | Divergence-free winds |
|----|---------------------------|----------------|----------|-------------|------------------------|
| 01 | Troposphere | 250 m | 4 h | 1,32 | No |
| 02 | PBL | 250 m | 4 h | 1,32 | No |
| 03 | PBL | 500 m | 4 h | 1,32 | No |
| 04 | PBL | 750 m | 4 h | 1,32 | No |
| 05 | PBL | 1/2 PBL | 4 h | 1,32 | No |
| 06 | PBL | 1/2 PBL | Kinetic equation[a] | Modeled[b] | No |
| 07 | PBL | 1/2 PBL | Kinetic equation[a] | Modeled[b] | Yes[c] |

(a) Lifetime computed using Eq 3 with $NO_2$-weighted average values of T and OH extracted from LOTOS-EUROS for the closest simulated hour before the application of the FDA, only the vertical levels below the PBL height were used. (b) Values for each pixel and day using the simulated NO and $NO_2$ VCD from LOTOS-EUROS. (c) divergence-free wind dataset generated using the iterative algorithm described by Bryan (2022).

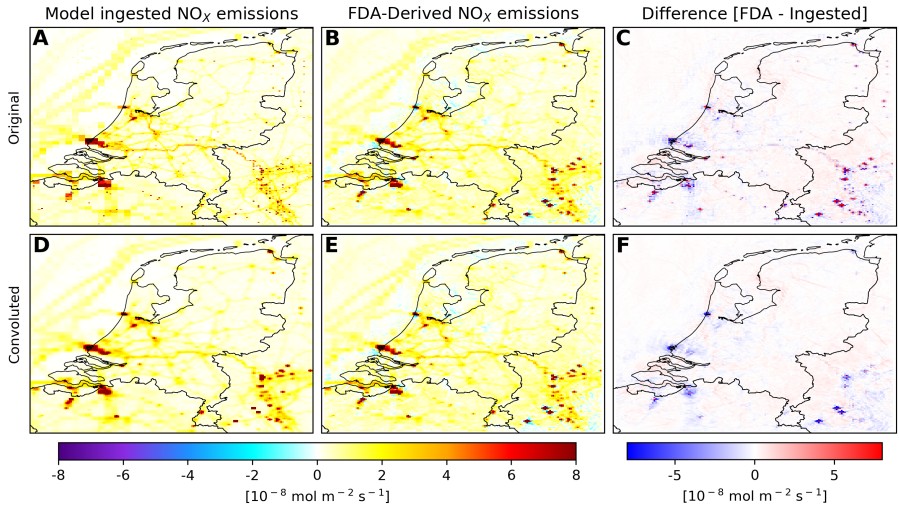

**Figure 3.** Comparison between the original and convoluted $NO_x$ model-ingested emissions and FDA-derived emissions for June, July, and August (JJA), at 13:30 LT, using configuration ID06. **(A)** Original model-ingested $NO_x$. **(B,E)** FDA-derived $NO_x$. **(C)** Difference between the FDA-derived $NO_x$ and the original ingested $NO_x$. **(D)** Convoluted model-ingested $NO_x$. **(F)** Difference between the FDA-derived $NO_x$ and the convoluted ingested $NO_x$

the original ingested data, the FDA underestimate $NO_x$ emissions for the hotspot cells and overestimate $NO_x$ emissions at the surrounding cells (Figure 3C). However, when compared to the convoluted version of the ingested emissions, there is a better agreement between the datasets (Figure 3F).

### 3.2.2 Performance evaluation based on simulated TROPOMI satellite $NO_2$ observations

To evaluate the performance of the FDA under satellite-like observation conditions the FDA was applied using the synthetic observations at the TROPOMI grid. This includes coarser horizontal resolution, the actual TROPOMI irregular tilted push-broom observation grid, and missing pixels due to clouds using the TROPOMI qa_value flag. Note that these are idealized observations because no observation noise was added. For this evaluation, the wind fields, lifetime, and the $NO_2$ to $NO_x$ conversion factor were selected based on the configuration that demonstrated optimal performance (minimizing NMB and NMGE) during the sensitivity analyses conducted in the preceding stage of this research.

### 3.2.3 Evaluations with real TROPOMI data

Finally, the FDA was used on real TROPOMI data, employing both the default TM5-MP $NO_2$ *a-priori* profile shape and the updated LOTOS-EUROS $NO_2$ profile, to assess the influence of the profile shape used in the retrieval on the emission estimate. Once more, the selection of wind fields, lifetime, and the $NO_2$ to $NO_x$ conversion factors were determined based on the best scenario identified during the sensitivity analyses.

### 3.3 Spatiotemporal interpolations and other data-processing considerations

Temporal interpolations for the various variables were conducted linearly using the two nearest simulated hours to the desired timestamp. Similarly, wind vertical interpolation followed a linear approach, considering the two nearest vertical levels of the LOTOS-EUROS model to the desired altitude. Chemical fields (NO, $NO_2$, and OH) were horizontally regridded when necessary using a conservative interpolation method, while meteorological fields (winds, PBL height, and temperature) were regridded using a bilinear interpolation method. We made use of the xESMF library (Zhuang et al., 2023) within the Python programming language.

To apply the FDA using synthetic observations at the TROPOMI grid alongside actual TROPOMI data, divergence and emissions are estimated on a tilted irregular grid for each orbit. Subsequently, the emissions are regridded into a regular 5x5 km array to facilitate averaging into the JJA mean emissions, utilizing the conservative interpolation method of xESMF once again. It is important to note that estimating the divergence on the TROPOMI grid requires the rotation of the wind fields from a east-west and north-south components into across-track and along-track components.

## 4 Results and discussion

### 4.1 FDA performance for model grid observations

The FDA was implemented in seven different experiments listed in Table 1. The performance evaluations, summarized in Table 2 and Figure 4, indicate that experiment ID06 offers the best NMB values for the entire domain (3,2%), low NMB over hotspot regions (-8,6%), and a standard deviation that aligns closely with the convoluted emissions input into LOTOS-EUROS. This indicates improved accuracy in capturing the variability of the JJA prior $NO_x$ emissions. Therefore, the results of the test ID06

**Table 2.** Performance metrics using the synthetic observations at LOTOS-EUROS grid

| ID | Total emissions [a] | Original ingested emissions | | | | | | Convoluted ingested emissions | | | | | |
|---|---|---|---|---|---|---|---|---|---|---|---|---|---|
| | | Entire domain | | | Hotspots | | | Entire domain | | | Hotspots | | |
| | | NMB | NMGE | R | NMB | NMGE | R | NMB | NMGE | R | NMB | NMGE | R |
| 01 | 981,7 | 59,0 | 98,6 | 0,71 | -8,6 | 46,4 | 0,86 | 59,0 | 74,4 | 0,95 | 16,5 | 25,9 | 0,98 |
| 02 | 567,8 | -8,0 | 63,2 | 0,74 | -35,9 | 47,8 | 0,87 | -8,0 | 42,1 | 0,95 | -18,3 | 23,9 | 0,98 |
| 03 | 589,7 | -4,5 | 61,0 | 0,75 | -32,6 | 46,1 | 0,88 | -4,5 | 38,9 | 0,96 | -14,0 | 21,5 | 0,98 |
| 04 | 592,8 | -4,0 | 63,4 | 0,75 | -31,7 | 46,8 | 0,89 | -4,0 | 41,2 | 0,96 | -12,9 | 22,5 | 0,97 |
| 05 | 589,6 | -4,5 | 60,4 | 0,76 | -31,6 | 45,3 | 0,89 | -4,5 | 37,8 | 0,97 | -12,8 | 20,0 | 0,98 |
| 06 | 636,8 | 3,2 | 63,2 | 0,80 | -28,3 | 43,7 | 0,92 | 3,2 | 42,3 | 0,94 | -8,6 | 22,3 | 0,96 |
| 07 | 638,8 | 3,5 | 61,8 | 0,80 | -28,8 | 43,7 | 0,92 | 3,5 | 41,0 | 0,94 | -9,2 | 22,3 | 0,96 |

**(a)** FDA-derived $NO_x$ aggregated emissions for the entire simulation domain in mol s$^{-1}$. Total *a-priori* $NO_x$ emissions used as input in LOTOS-EUROS were 617.4 mol s$^{-1}$.

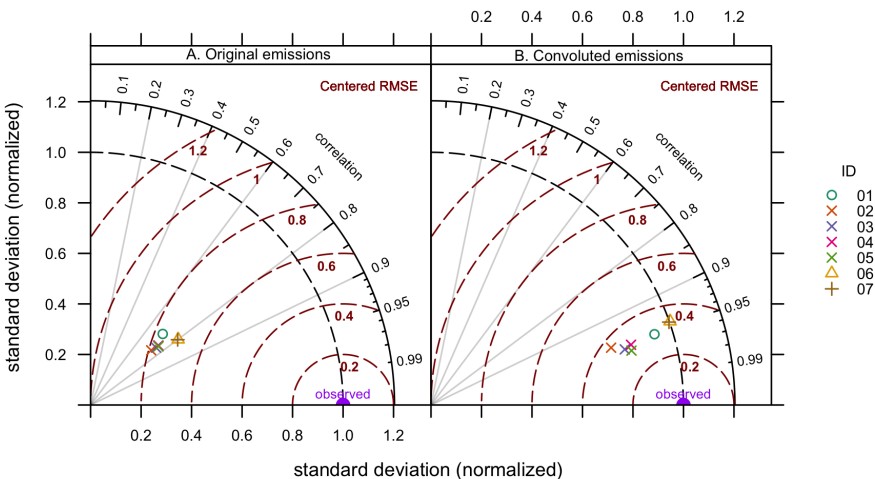

**Figure 4.** Taylor diagram summarizing the FDA performance for the seven sensitivity test of table 1 with ID 01 to 07 using **(A)** the original model-ingested emissions and **(B)** convoluted model-ingested emissions as a reference

are used as a benchmark for the following discussion. Note that test ID07, where the wind divergence is mitigated, achieves similar performance metrics.

Fig 5 compares the JJA mean $NO_x$ emissions ingested into LOTOS-EUROS and those derived with the FDA. Overall, the FDA effectively reconstructs the spatial distribution of emissions, capturing major industrial and urban sources such as the port of Rotterdam, the city of Amsterdam, and Schiphol Airport. Moreover, the FDA captures minor emission sources, exemplified by the retrieved emission in the road network, inland shipping along the Rhine River, and small hotspots in Den Helder, Leeuwarden, and Groningen, located in the northern region of the Netherlands.

Despite the good correspondence in spatial distribution, the FDA spread out the emission hotspots to neighboring grid cells, particularly evident in the Ruhr area. This is caused by numerical smearing which induces an underestimation of emissions at the hotspots, misallocating part of the emissions to neighboring cells (Cooper et al., 2017). This smearing arises from the numerical solution of the divergence term which is unable to represent sharp fluctuations with a size of the order of a single grid box. The fourth-order central-finite difference employed in this research incorporates data from adjacent grid cells in both east-west and north-south directions, thereby smoothing the divergence term. The smearing effect may also result from biases in the CTM-predicted $NO_2$ fields due to numerical diffusion when resolving the advection equation at the model grid, which artificially smooths the solution in regions with steep concentration gradients, leading to plume stretching (Eastham and Jacob, 2017; Rastigejev et al., 2010).

The impact of pollutant transport on emission smearing was not discernible in the findings of this study. Figure A2 illustrates emission maps grouped by days predominantly experiencing northerly and southerly winds. Emissions exhibit smearing in all directions surrounding the hotspots, attributed to the aforementioned numerical effects, rather than being primarily directed towards cells in the downwind direction. TROPOMI measures $NO_2$, which is not immediately formed at the source, but only after the reaction of NO with $O_3$ has taken place. Consequently, a displacement downwind of the source location occurs as NO converts into $NO_2$ and becomes visible to the satellite. This did not occur in the synthetic experiments presented here because the $NO_2$ to $NO_x$ ratio was derived from modeling outputs, ensuring that all atmospheric $NO_x$ was included.

Regarding total emission values, the FDA underestimates emissions in hotspot areas, with an average NMB of -8,6%. This discrepancy is primarily due to the smearing effect previously discussed, which is not entirely corrected when compared to the convoluted emissions ingested by the model. Furthermore, inaccuracies in representing the lifetime of $NO_x$ will contribute to the bias. Indeed, the modeled OH concentrations, used in Eq 3, are highly uncertain due to their numerous sinks, rapid cycling, and nonlinear chemical feedbacks. Discrepancies in OH simulations across different models can be considerable, as shown in Figure A3, owing to differences in photolysis rates, cloud parameterizations, radiative transfer codes, and the representation of volatile organic compounds within each model (Nicely et al., 2017, 2020). These factors collectively influence OH chemistry and, consequently, its predicted concentrations. Furthermore, OH concentrations are also affected by the resolution of the model due to nonlinear effects. Higher resolution simulations exhibit an enhanced titration of OH with $NO_x$ in $NO_x$-saturated regions, leading to an extended lifetime, while having the opposite effect in $NO_x$-limited regions (Li et al., 2023; Krol et al., 2024). In contrast to the hotspots, the FDA slightly overestimates background emissions (NMB of 3,2%). This is mainly associated with biases in the lifetime estimation. Likewise, negative emission values may occur due to biases in the chosen wind fields, affecting the estimation of divergence.

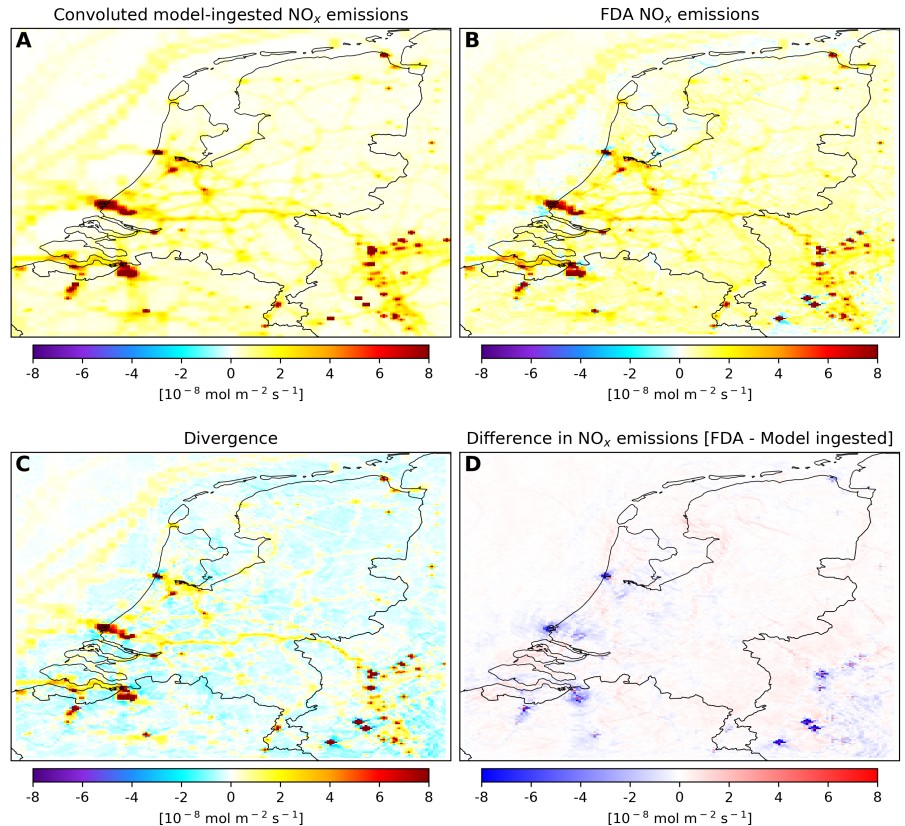

**Figure 5.** Mean $NO_x$ emissions and divergence for June, July, and August (JJA) at 13:30 LT derived from the synthetic dataset at LOTOS-EUROS native resolution using configuration ID06. **(A)** Convoluted $NO_x$ emissions ingested into LOTOS-EUROS. **(B)** FDA-derived $NO_x$ emissions. **(C)** $NO_x$ flux divergence. **(D)** Difference in $NO_x$ emissions between the FDA-derived dataset and the convoluted model-ingested emissions.

In summary, the FDA demonstrates a good performance in reconstructing the spatial pattern of emissions and maintaining a low bias in hotspot areas provided that wind, $\tau$, and the $NO_2$ to $NO_x$ conversion factors are known to good accuracy.

### 4.1.1 Column integration height

Experiments ID01 and ID02 examine the effect of the integration height of the $NO_2$ column used for emission calculations. This test has the most significant impact on the overall results. Using the full modeled tropospheric column (Test ID01) results in an overestimation of emissions by 59% across the entire domain. In contrast, integrating the column only up to the PBL height leads to a slight underestimation of -8%. The integration height of the column primarily affects the sink term in equation 1, as it is directly proportional to the $NO_x$ column. Using the full tropospheric column leads to a significant overestimation

because residual $NO_x$ from long-range transport above the PBL top or entrainment with the upper boundary conditions of

the simulation is included. According to our simulations, the free tropospheric $NO_2$ can represent between 20 to 65% of the column. This residual $NO_x$ has no direct relation to the instantaneous emissions being evaluated at the time the FDA is applied. Additionally, the PBL and free tropospheric $NO_2$ are exposed to different chemical sinks. Thus, using a single lifetime value of 4 hours may not accurately represent the entire tropospheric column.

The residual $NO_2$ column obtained after subtracting the PBL contribution (See Figure A4), still exhibits significant concentrations over the Ruhr area, Antwerp, and along the Dutch coastline. These regions contain the majority of $NO_x$ emissions, suggesting that the spatial patterns in the residual layer may reflect enhancements in the free troposphere caused by emissions from previous hours. The high concentration in the residual layer can also originate because the PBL does not align precisely with the model layer interfaces, leaving part of the column excluded from the inversion process. This can partially explain the 385   low negative bias of -8% when deriving emissions using this approach.

     A similar approach was impelementend by Liu et al. (2021), who used the FDA to derive methane emissions. In their study, they used a PBL column of methane and validated this approach against GEOS-Chem outputs. Their validation also demonstrated improved performance of the FDA, as using a PBL column excludes transport in the upper troposphere. Furthermore, Koene et al. (2024) noted that removing background fields is advantageous because it allows the analysis of wind fields to be 390   confined to the PBL rather than the entire column, and also eliminates the need to make assumptions about the steady-state conditions for the background field.

### 4.1.2   Wind fields

Experiments from ID02 to ID05 were conducted with all variables held constant, except for varying the altitude of the chosen 2D wind field. The results of these experiments show similar performance in reproducing the original emission dataset. The 395   total emissions obtained differ only by 4% and correlation and NMGE exhibit comparable values, as summarized in Table 2. Additionally, the mean divergence maps, depicted in Fig A5, reveal close agreement regardless of the chosen wind altitude. These results may be specific to the study area, given the uniformity of wind patterns in the Netherlands due to the flat terrain. Indeed, as depicted in Figure A6, wind speed and direction over land exhibit minimal variations with altitude, with the primary distinction being an increase in wind speed near the Netherlands coast when using the half PBL height for selecting the wind 400   fields. Application of the FDA in areas with more complex orography may require additional investigation into selecting the optimal wind height.

     Despite the observed similarities, it is still advisable to dynamically select the wind field altitude as a function of the PBL to capture its day-to-day variability and spatial patterns such as contrasts between land and sea. As demonstrated in Fig A7, this variability is evident, with PBL values fluctuating between altitudes lower than 100 m and higher than 1500 m. In this study, 405   using a half PBL altitude for the wind fields provided the best metrics for NMB (-12,8%), NMGE (20,0%), and correlation (0,98) for the hotspots in comparison to the fixed wind altitude alternatives.

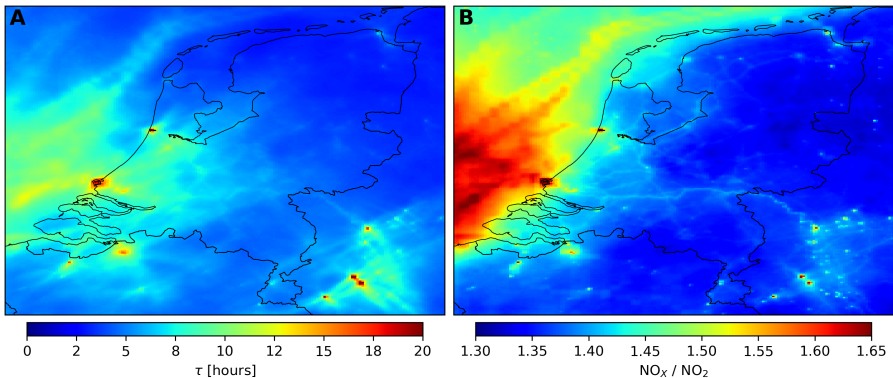

**Figure 6.** Mean $NO_2$ lifetime ($\tau$) and $NO_x$ to $NO_2$ ($L$) for June, July, and August (JJA) derived from LOTOS-EUROS simulation, at 13:30 LT. **(A)** Mean PBL $NO_2$-weighted $\tau$ determined using Equation 3, employing average values of temperature (T) and hydroxyl radical (OH) from LOTOS-EUROS for the two hours immediately preceding the application of the FDA. **(B)** Mean L, determined by the ratio between vertical column densities of $NO_x$ and $NO_2$ from LOTOS-EUROS

### 4.1.3 $NO_2$ lifetime and $NO_2$ to $NO_x$ conversion factor

Using a lifetime derived from OH and $T$ and a $NO_2$ to $NO_x$ conversion factor from the LOTOS-EUROS simulation (Test ID06) yields notable enhancements compared to employing single fixed values across the entire period and domain (Test ID05). Specifically, this approach diminishes the underestimation over hotspot regions (-8,6% vs -12,8%), and improves the bias over the entire domain in general (3,2% vs -4,5%). Furthermore, the normalized standard deviation depicted in Figure 4, indicates that the variability of the derived emissions with this method more closely matches the variability of the *a-priori* $NO_x$ emissions ingested in LOTOS-EUROS. Note that while the results using a fixed lifetime of 4 h are still acceptable, the method heavily depends on the assumed value. For example, changing the assumed value to 5 h would increase background underestimations to -20%, whereas using 3 h would result in an overestimation of 22%.

The mean lifetime map, shown in Figure 6A, reveals a strong spatial variability at the spatial resolution of the model. In areas with high emission levels, lifetimes surpass 20 hours, reflecting the low local OH concentrations. Similarly, lifetimes over the North Sea range from 8 to 14 hours, also due to a reduced availability of OH which is depleted with fresh shipping emissions. Both scenarios show a substantial departure from the 4-hour estimate cited by Beirle et al. (2019). In the North-East part of the Netherlands however, emission density is lower and the lifetime becomes closer to the 4-hour value. Using modeling-derived lifetime better represents ageing of the pollution plumes. Figure A8 shows that the $NO_2$ lifetime reaches its maximum over the concentration hotspots and gradually decreases downwind according to LOTOS-EUROS. At the emission source, the lifetime peak due to local titration of OH, which is recovered downwind as the plume mixes with fresh air (Krol et al., 2024; Vinken et al., 2011). Considering the spatiotemporal variability of the $NO_2$ lifetime contributes to a more precise reconstruction of the original emissions in comparison to using a single lifetime value.

Due to the long lifetimes at the hotspots, the contribution of the $S$ term in Eq 1 becomes small, thus making $D$ the dominant factor for retrieving emissions at the hotspot location. On the contrary, for retrieving emissions at the country level or the entire simulation domain, the sink term is the dominating factor for estimating emissions. Indeed, if the net inflow or outflow of pollutants across the boundaries is negligible, the $D$ term becomes zero. Consequently, emissions are dominated by $S$, which is proportional to $1/\tau$, so a bias in $\tau$ will directly result in a bias in the total emission. This aspect represents a drawback of the method due to the considerable uncertainty of this parameter.

The complexities involved in estimating lifetimes are not unique to the FDA. Model-derived lifetimes, based on OH availability, can be biased, as previously discussed in Section 4.1. Alternative methods for deriving lifetimes include the simultaneous estimation of emissions and lifetimes using downwind $NO_2$ patterns derived from the satellite observations directly. This approach reduces reliance on prior assumptions or model-based inputs and is applied in plume dispersion methods to estimate emissions from single (Beirle et al., 2011; Valin et al., 2013) or multiple sources (Fioletov et al., 2022; Dammers et al., 2024). However, this method is not always feasible, as plumes are not always isolated and cannot always be idealized as point sources. Additionally, covariance between emissions and lifetimes introduces errors into lifetime estimations.

Figure 6B illustrates the mean $NO_x$ to $NO_2$ ratio simulated by LOTOS-EUROS. Deviations from the value of 1,32 proposed by Beirle et al. (2019) manifest prominently over concentration hotspots, the North Sea, and adjacent coastal regions, where elevated ratios reaching up to 1,60 are obtained. The higher ratios over the hotspot areas are caused by the proximity to large concentrations of freshly emitted NO, leading to local titration of $O_3$; hence, reducing the reactions to form $NO_2$. Similarly, the presence of ship emissions over the North Sea and coastal regions, along with a shallower boundary layer compared to inland areas, contributes to the reduction of $O_3$ in the marine boundary layer and accounts for the high ratios.

As previously discussed for the lifetime, deriving a $NO_2$ to $NO_x$ conversion factor from simulation outputs can be advantageous for estimating emissions over extended periods. Alterations in atmospheric chemical processes, driven by different emission scenarios, can influence the non-linear relationship between $NO_x$ and $O_3$, thereby altering the $NO_x$ to $NO_2$ ratio. In the Netherlands scenario, Zara et al. (2021) observed a decrease in NO-titration from 2005 to 2018 due to lower $NO_x$ emissions, causing a shift in the NO–$NO_2$ equilibrium towards higher concentrations of $NO_2$. This led to the reductions in the surface air $NO_x$ to $NO_2$ ratio during winter (from 1,45 to 1,28) and summer seasons (from 1,30 to 1,20).

### 4.1.4 Divergence-free winds

For the period and domain used in this study, a large-scale wind divergence primarily manifests along the coastline, as exemplified in Figure A9. The iterative algorithm used to mitigate the wind divergence in the dataset (Bryan, 2022) successfully diminishes it, without substantially altering the wind patterns. Using these divergence-free wind datasets (Test ID07) does not substantially change the retrieved emissions, resulting in only a slight reduction in the NMGE (42,3% vs. 41,0%) for the entire domain. The other performance metrics remain almost identical, indicating that the impact of wind divergence in the domain and period studied is small. The small effect of reducing the wind divergence term in our findings is due to our integration of $NO_2$ VCD only up to the PBL height. As noted by Koene et al. (2024), addressing the wind divergence term can be beneficial when the background is not been subtracted from the columns.

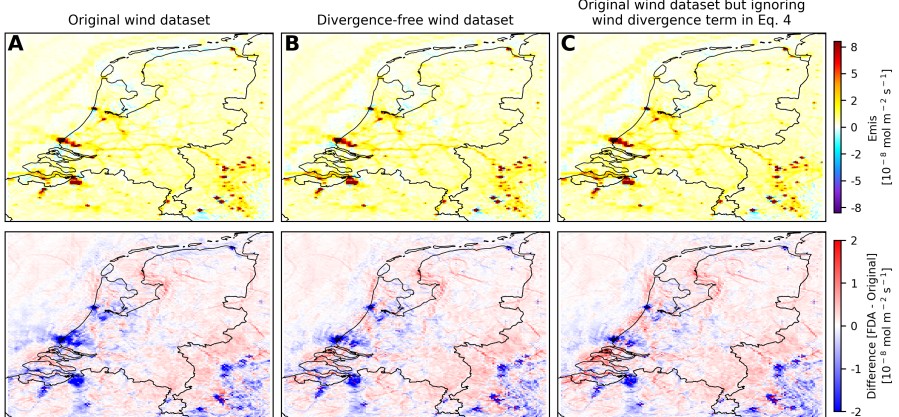

**Figure 7.** Comparison of mean $NO_x$ emissions for June, July, and August (JJA) at 13:30 LT derived from the synthetic dataset at LOTOS-EUROS native resolution using the **(A)** original wind dataset, **(B)** the divergence-free wind dataset, and **(C)** original wind dataset but skipping the divergence term in Eq 4, and difference map between the derived FDA-emissions and the emissions originally ingested into LOTOS-EUROS.

Note that the flux divergence in Eq 1 can be expanded using the product rule into advection and wind divergence terms,

$$D = \nabla(LV\mathbf{w}) = \mathbf{w} \cdot \nabla(LV) + LV \cdot \nabla(\mathbf{w}) \tag{4}$$

Beirle et al. (2023) suggested omitting the wind divergence term, arguing that its influence is insignificantly small for D. However, Figure 7C, shows that by excluding the divergence term in Eq 4 there is an increased emission overestimation at the coastline.

### 4.1.5 Application of the FDA for other times of the day

In addition to the TROPOMI overpass time test, two additional scenarios were examined at 08:00 and 18:00 LT. This was conducted to gain insights into the applicability of the FDA beyond stable conditions, which could have implications for employing the method to derive emissions from geostationary satellites such as GEMS, TEMPO, and the upcoming Sentinel-4 sensor. The same experimental settings as in ID06 were employed for this assessment.

Figure 8 shows the derived emission and divergence maps for the scenarios above. At 08:00 LT, the emissions and divergence map are almost equal. A low concentration of OH during this period leads to significantly longer lifetimes computed with Eq 3, thus the sink term in the Eq 1 becomes negligible. This leads to extensive negative emission areas, as revealed in Figure 8B. These errors mainly stem from ignoring the accumulation of atmospheric $NO_x$ during non-stable conditions, within a PBL in development and with changing emissions. Therefore, deviating from the steady-state assumption in the FDA. Additionally, because the chemical lifetime is significantly extended due to reduced photochemistry at this time, the sinks due to deposition, previously unaccounted, may become more significant and need to be included to compensate for the negative emission artifacts.

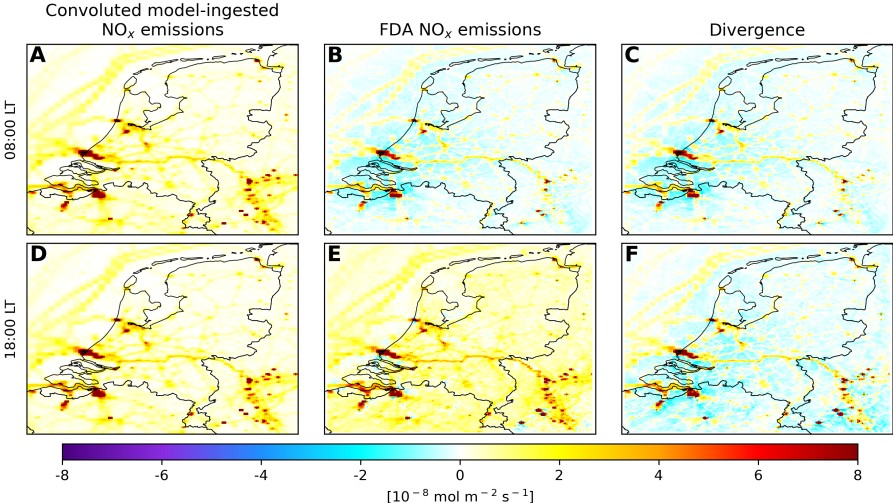

**Figure 8.** Mean $NO_x$ emissions and divergence for June, July, and August (JJA) derived from the synthetic dataset at LOTOS-EUROS native resolution using configuration ID06, at 08:00 and 18:00 LT. **(A)** Convoluted $NO_x$ emissions ingested into LOTOS-EUROS at 08:00 LT. **(B)** FDA-derived $NO_x$ emissions at 08:00 LT. **(C)** $NO_x$ flux divergence at 08:00 LT. **(D)** Original $NO_x$ emissions ingested into LOTOS-EUROS at 18:00 LT. **(E)** FDA-derived $NO_x$ emissions at 18:00 LT. **(F)** $NO_x$ flux divergence at 18:00 LT.

At 18:00 LT, the emission map displays no negative artifacts (Figure 8E), but emissions are overestimated across the entire domain by 24,8% in average. The biases can be attributed to uncertainties in the diurnal variations of the lifetime and chemistry. During this time of day, photochemical processes diminish, making it necessary to consider additional competing reactions that consume OH and other sink pathways for $NO_x$ besides the formation of $HNO_3$.

The diurnal profile of $NO_x$ and $NO_2$ VCD within the PBL, derived from the LOTOS-EUROS simulation, undergoes distinct shifts throughout the day as shown in Fig. 9A. **(1)** During the period from 19:00 to 05:00 LT, $NO_x$ is consumed through titration reactions with $O_3$, forming $NO_3$, which can subsequently react with another $NO_2$ molecule to produce $N_2O_5$. With low emissions of $NO_x$ during this time, there is an insufficient supply to compensate for the consumed atmospheric $NO_x$, resulting in a decrease in its concentration. **(2)** Between 05:00 and 12:00 LT, there is an increase in $NO_x$ emissions. The $O_3$ levels remain low following the night-time titration, which diminishes the availability of OH required to consume $NO_2$ and form $HNO_3$, consequently resulting in an accumulation of $NO_x$. **(3)** From 11:00 to 16:00LT, increased photochemistry and entrainment of free tropospheric $O_3$ into the boundary layer elevates $O_3$ concentrations, thereby increasing the production of OH, which subsequently consumes $NO_2$, leading to a reduction in $NO_x$. Furthermore, the PBL reaches its peak during this time, enhancing the mixing of $NO_X$ and VOCs, which further promotes OH formation. **(4)** As the day progresses from 16:00LT to 19:00LT, photochemistry diminishes, with $O_3$ being primarily used in the conversion of NO to $NO_2$. The availability of OH to consume $NO_2$ decreases, resulting in the accumulation of $NO_x$ levels.

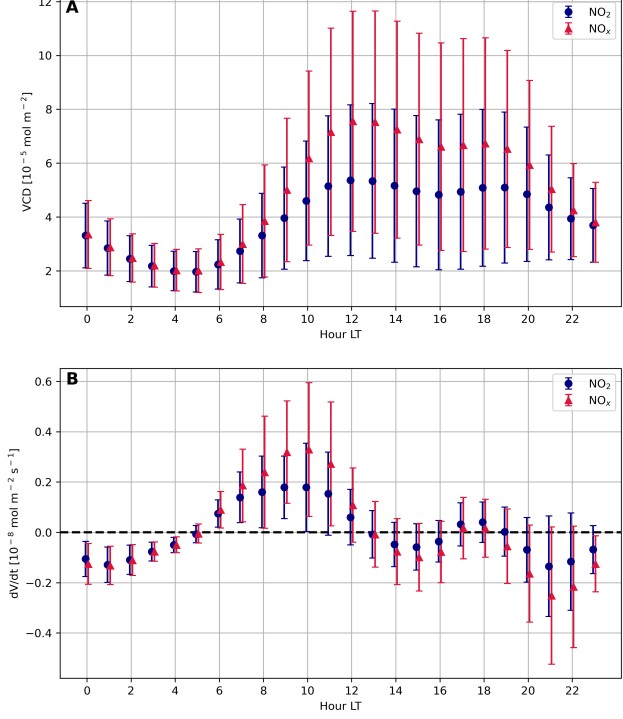

**Figure 9.** Mean diurnal profile for the entire simulation domain of $NO_2$ and $NO_x$ vertical column densities (VCD) within the PBL for June, July, and August (JJA) derived from LOTOS-EUROS simulation **(A)**, and change rate of the VCD considering the hours $i$ and $i-1$ **(B)**. The vertical lines represent the standard deviation of each point.

The fluctuations in $NO_x$ VCD over time ($dV/dt$), as illustrated in Fig. 9B, indicate that relying on the assumption of steady-state in the FDA is insufficient at many hours of the day. Including this term will transform Eq. 1 into

$$E = D + S + \frac{dV}{dt} \tag{5}$$

thereby mitigating the underestimations observed when deriving emissions at 08:00 LT (Fig. 8B), as a positive term will be introduced to compensate for $NO_x$ buildup. In addition, accounting for deposition-related sinks might be necessary in the early morning, as their contribution becomes significant due to the lower photochemistry during that period. It is worth noting that estimating and integrating this term is feasible in this scenario due to the availability of hourly VCD data from the simulation, and could be extended to geostationary satellite data with hourly observations. However, for polar orbit satellites like TROPOMI, evaluating this term is not viable for a single instrument, as observations are limited to just one per day. Nonetheless, it would be possible to estimate changes in $NO_2$ VCD using multiple polar-orbiting satellites with similar overpass times. This approach has been demonstrated by Penn and Holloway (2020) using the GOME-2 and OMI instruments, as well as by Boersma et al. (2009) using SCIAMACHY and OMI.

For the scenario at 18:00LT (Fig. 8E), incorporating the $dV/dt$ term will not significantly affect the derived emissions as $dV/dt$ is small at this time of the day. For an understanding of the overestimation in this scenario it is needed to reassess the lifetime estimates. This discrepancy might stem from uncertainties in the OH estimates utilized for deriving the lifetime, or it could indicate the need to incorporate additional competing reactions that consume OH and other sink pathways for $NO_x$, such as oxidation to produce alkyl and multifunctional nitrates (Sobanski et al., 2017; Romer Present et al., 2020). In addition, as photochemical processes diminish during this time of day, the contribution of $NO_x$ loss due to deposition could become more significant.

## 4.2  FDA performance for synthetic TROPOMI observations

The accuracy of the FDA was further assessed under satellite-like observing conditions, employing synthetic observations sampled at the TROPOMI grid using TROPOMI quality screening. To conduct this evaluation, the configuration used in experiment ID06 was employed (See Table 1), as they yielded the best performance for the test conducted over the synthetic observations at the native LOTOS-EUROS grid.

The results from this test show that despite the lower resolution, irregular grid, and presence of missing pixels due to cloud cover and retrieval issues, the FDA is capable of reconstructing the spatial distribution of emissions, particularly for major emitters as shown in Figure 10. However, discerning minor emission sources like roads, inland shipping, and smaller industries proves challenging at this lower resolution. The underestimation of hotspot emissions persists for this test (NMB: -11,0%; NMGE: 18,5% with respect to the convoluted model-ingested emissions), as well as the overestimation of background regions (NMB: 17,7%; NMGE: 39,2%). These biases may be attributed to inaccuracies in the $NO_x$ lifetime and the wind fields, as discussed previously in this study.

Note also that the generalized overestimation of the background increased from 3,2% in the synthetic observations at the model grid to 17,7% in the synthetic observations at the TROPOMI grid. This increase can partly be attributed to a sample bias, as only the pixels with cloud-free conditions are considered in this later scenario to emulate TROPOMI observations. By focusing solely on cloud-free conditions, temperature-dependent emissions in the model are amplified, impacting not only $NO_x$ emissions but also other compounds that react with $NO_x$ in the atmosphere, thereby affecting its lifetime. In LOTOS-EUROS, these temperature-dependant emissions include $NO_x$ emissions from soils, biogenic VOC (also influenced by photosynthetically active radiation), and road transport emissions (Manders et al., 2021). Additionally, the increase in incoming solar radiation under cloud-free conditions enhances the photolysis of several compounds. Furthermore, the number of observations per month or pixel can vary, affecting the emissions derived from the inversion process.

## 4.3  Real TROPOMI data

Previous studies have shown that replacing the TROPOMI coarse TM5-MP $NO_2$ *a-priori* vertical profile with a high-resolution profile offers a more accurate representation of meteorological and chemical fields which increase column concentrations near emission sources by more than 30% and create a steeper concentration gradient around these areas (Laughner et al., 2016; Griffin et al., 2019; Zhao et al., 2020; Judd et al., 2020), leading to better agreement with ground-based validation (Douros

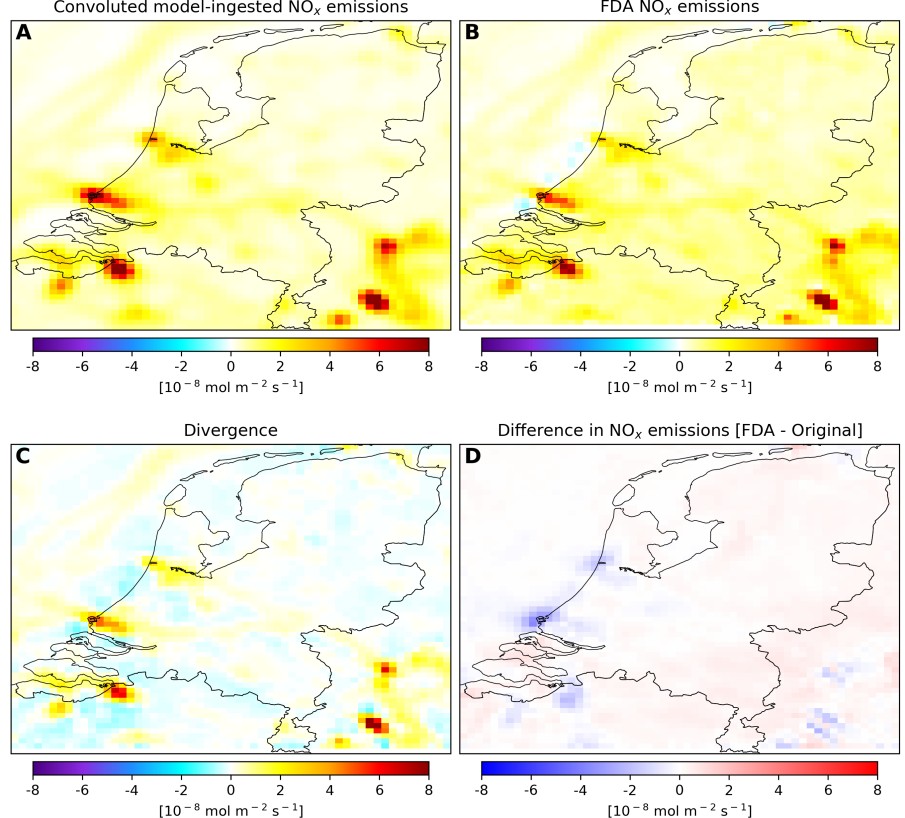

**Figure 10.** Mean $NO_x$ emissions and divergence for June, July, and August (JJA) derived from the synthetic dataset sampled at TROPOMI grid and timestamp using configuration ID06. **(A)** Convoluted model-ingested $NO_x$ emissions. **(B)** FDA-derived $NO_x$ emissions. **(C)** $NO_x$ flux divergence. **(D)** Difference in $NO_x$ emissions between the FDA-derived dataset and the convoluted model-ingested emissions.

et al., 2023; Liu et al., 2020; Ialongo et al., 2020). Based on these findings, we conducted two tests using TROPOMI with the
default TM5-MP $NO_2$ *a-priori* profile shape, and a modified version incorporating an updated LOTOS-EUROS $NO_2$ profile. These observations represent the total tropospheric VCD, which are subsequently converted to PBL VCD by multiplying by the ratio of PBL VCD to tropospheric VCD, as derived from LOTOS-EUROS simulations. This approach allows for a consistent methodology for real TROPOMI data, matching the one employed for the synthetic observations in experiment ID06.

    Figure 11 illustrates the contrast in $NO_2$ VCD measurements between the TROPOMI TM5-MP product and the LOTOS-
EUROS *a-priori* replacement version. The modified version of TROPOMI with LOTOS-EUROS *a-priori* exhibits notably higher $NO_2$ VCD values, averaging a 23% increase concerning the default TM5-MP *a-priori* product. This difference directly impacts the emission estimate, resulting in total emissions derived from TROPOMI LOTOS-EUROS being 15% higher on average, which increases to 34% when considering only the hotspot pixels. These results align with earlier research referenced previously. The updated *a-priori* profile not only enhances $NO_2$ vertical column densities (VCDs) at hotspot locations but also

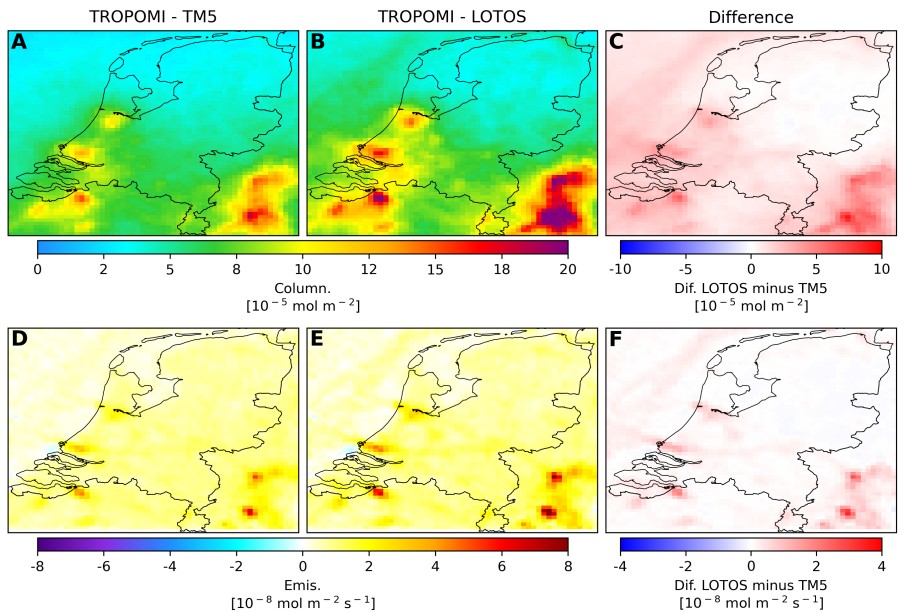

**Figure 11.** Mean $NO_2$ vertical column densities (VCD) and $NO_x$ emissions for June, July, and August (JJA) derived using real TROPOMI data, with default TM5-MP and high-resolution *a-priori* profile shape. **(A)** TROPOMI $NO_2$ VCD with default TM5-MP *a-priori*. **(B)** TROPOMI $NO_2$ VCD with LOTOS-EUROS *a-priori*. **(C)** Difference in $NO_2$ VCD between TROPOMI products. **(D)** FDA-derived $NO_x$ emissions using TROPOMI with default TM5-MP *a-priori*. **(E)** FDA-derived $NO_x$ emissions using TROPOMI with LOTOS-EUROS *a-priori*. **(F)** Difference in $NO_x$ emissions between TROPOMI products

provides a more accurate representation of concentration gradients, which impacts the estimation of the divergence term in the FDA (Laughner et al., 2016). Additionally, because our study uses consistent wind data for both generating the new *a-priori* profiles and calculating the divergence, we expect more accurate divergence estimates.

  A comparison between the convoluted bottom-up emission inventory used in the model and the one derived using TROPOMI with LOTOS-EUROS *a-priori*, as presented in Figure 12, shows a good agreement between the datasets. The correlation coef-
555 ficient is 0,83 for the entire domain and 0,79 for the hotspot areas. Using the bottom-up inventory as the baseline, TROPOMI-derived emissions are 32,2% higher for the entire domain, potentially due to TROPOMI signal issues, non-negligible accumulation terms ($dV/dt$), or differences in soil emissions. On the contrary, when considering only hospot areas, emissions are biased low by 18%. Applying the FDA to the synthetic data (see section 4.2) revealed an 11% negative bias at hotspot locations, indicating a general tendency of the FDA to underestimate emissions at these points. The additional 7% bias observed when
using real TROPOMI data with the updated *a-priori* falls within the uncertainty ranges of both TROPOMI and bottom-up inventories.

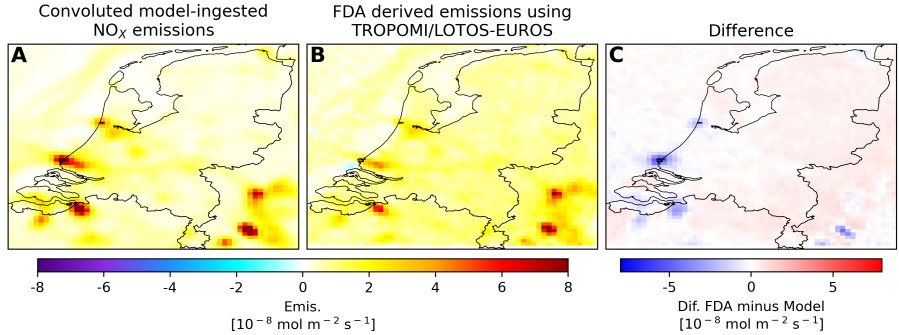

**Figure 12.** Mean $NO_2$ vertical column densities (VCD) and $NO_x$ emissions for June, July, and August (JJA) derived using real TROPOMI data, with default TM5-MP and high-resolution *a-priori* profile shape. **(A)** Convoluted model-ingested $NO_x$ emissions. **(B)** FDA-derived $NO_x$ emissions using TROPOMI with LOTOS-EUROS *a-priori*. **(C)** Difference in $NO_x$ emissions between the model-ingested and FDA-derived datasets.

## 4.4 General performance of the method

The application of the FDA on synthetic data at 13:30 LT demonstrates excellent results. There is a relatively low underestimation at hotspots and a slight overestimation of overall emissions for the entire domain by 3%, as summarized in Table 3. These results, however, are not achieved with the basic FDA implementation. Instead, they require the use of $NO_2$ VCD integrated up to PBL height, along with dynamically selected wind fields, $NO_2$ lifetime, and $NO_2$ to $NO_x$ conversion factors. Ignoring these assumptions and relying on tropospheric VCD and fixed values for the aforementioned factors increases the generalized overestimation to an average of 59%. The favorable results with the optimal settings also suggest that the steady-state assumption at this time of day is valid, as $dV/dt$ is sufficiently small (see Figure 9), allowing the FDA to be applied under standard noon conditions.

Conversely, when applying the FDA at different times of the day, which is necessary for handling geostationary observations, the steady-state assumption does not hold for several periods. For instance, performance using synthetic data at 08:00 LT shows extensive negative emission artifacts. This issue can be mitigated by including the $dV/dt$ term, as suggested in Eq 5, to compensate for the depletion of atmospheric $NO_x$ during this time. The performance of the FDA when analyzing synthetic data at 18:00 LT is significantly better, as the $dV/dt$ term is notably small at this time of day. However, there is still an overestimation for the entire domain of 25%, which may be linked to biases in the lifetime estimation.

The FDA performance at noon using synthetic data in the actual TROPOMI grid and timestamp also shows favorable results. However, the overestimation for the general domain increases from around 3% with synthetic data at the model grid to 18% with the TROPOMI grid. It is worth noting that this latter approach has a sampling bias, as pixels with cloud coverage were removed to mimic actual TROPOMI observations. Leaving only data from clear sky conditions can induce bias as all temperature dependant emissions in the model are enhanced under these circumstances. Moreover, the number of observations per month or pixel can vary, influencing the emissions obtained from the inversion

**Table 3.** Summary of performance metrics using configuration ID06 for the several test made using the convoluted model-ingested emissions as a reference

| Grid type | Test description | Convoluted ingested emissions | | | | | |
| | | Entire domain | | | Hotspots | | |
| | | NMB | NMGE | R | NMB | NMGE | R |
|---|---|---|---|---|---|---|---|
| Model | Synthetic data at 13:30 LT | 3,2 | 42,3 | 0,94 | -8,6 | 22,3 | 0,96 |
| Model | Synthetic data at 08:00 LT | -95,5 | 95,9 | 0,75 | -54,3 | 54,5 | 0,79 |
| Model | Synthetic data at 18:00 LT | 24,8 | 48,1 | 0,95 | 2,1 | 20,8 | 0,96 |
| TROPOMI | Synthetic data at TROPOMI timestamp | 17,7 | 39,2 | 0,82 | -11,0 | 18,5 | 0,91 |
| TROPOMI | TROPOMI-TM5 *a-priori* | 14,8 | 60,7 | 0,76 | -39,5 | 40,3 | 0,73 |
| TROPOMI | TROPOMI-LOTOS *a-priori* | 32,3 | 56,7 | 0,83 | -18,0 | 23,6 | 0,79 |

Lastly, it was possible to retrieve an emission dataset from actual TROPOMI observations that showed good agreement with the model-ingested emissions after replacing the TROPOMI TM5 *a-priori* with the high-resolution *a-priori* derived from LOTOS-EUROS. Emission hotspots were underestimated by -18%, while the overall domain showed an overestimation of 32%. This represents a good level of agreement, considering that the top-down satellite and bottom-up inventory emissions are fully independent pieces of information (observed versus reported emissions), and that the comparisons are done in a consistent way through the *a-priori* replacement.

The high-resolution model, incorporating prior knowledge of emission locations and strengths, provides important constraints on the spatial and temporal variability of $NO_2$ needed for quantitative comparisons. When bottom-up emissions significantly deviate from the real emissions, such as in cases of missing sources, this can result in errors in satellite-derived emissions of up to approximately 30%. These discrepancies can be reduced through iterative refinement, using satellite-derived emissions to update the model employed for generating the a *a-priori* estimates.

## 5 Conclusions

This study examined the FDA accuracy in reconstructing $NO_x$ emissions by utilizing satellite-synthetic observations generated with a high-resolution 2x2 $km^2$ implementation of the LOTOS-EUROS CTM over the Netherlands. Overall, the FDA demonstrates good performance, achieving a satisfactory representation of the spatial variation of emissions, particularly from major emitters (R: 0,96), while maintaining relatively low levels of underestimation for emission hotspots (NMGE: -8,6% in the LOTOS-EUROS grid and -11,0% in the TROPOMI grid). However, this level of agreement is only achieved by using $NO_2$ VCD integrated up to PBL height, dynamically selecting wind fields, $NO_2$ lifetime, and $NO_2$ to $NO_x$ conversion factors, all determined from the CTM output. This contrasts with the basic implementation of the method for actual satellite data, which in most cases relies on the total tropospheric column and fixed values for the aforementioned variables.

The sensitivity analyses provided the following insights. Using PBL $NO_2$ columns to drive the FDA results in a more accurate representation of emissions, reducing the generalized overestimation that occurs when using the tropospheric column. This improvement is because the entire tropospheric column includes $NO_x$ from long-range transport and upper troposphere entrainment, which are not directly linked to the emissions being assessed during the FDA application. Additionally, chemical sinks within the PBL and the free troposphere operate differently.

Using 2D wind fields at different altitudes exhibited a minimal impact on the final emission estimate. This is potentially attributed to the flat terrain and relatively smooth wind patterns over the Netherlands and the consistency of these winds within a well-developed PBL, which is the case for the summer period and noon overpass times that were used in this study. In this study, ECMWF-IFS wind fields, with an original resolution of roughly 9 km, were downscaled to a high spatial resolution of about 2 x 2 km. Nonetheless, the FDA is often applied using coarser publicly available datasets, such as ERA5, with a horizontal resolution of 0,25 x 0,25 degrees. Such coarse datasets may not accurately capture the spatial variability of wind fields, especially in regions with complex topography, which can introduce artifacts in divergence estimation. In these scenarios, using higher-resolution wind fields is recommended.

On the contrary, incorporating the spatiotemporal variability of $NO_x$ lifetimes and the conversion factors from $NO_2$ to $NO_x$ using modeling data emerged as a crucial factor affecting the precision of $NO_x$ emission derived from satellite data (Krol et al., 2024). The high-resolution model-derived lifetimes span a wide range of values, from as low as 2 hours in background conditions to over 20 hours over major sources, significantly diverging from the constant value of 4 hours proposed in the original FDA implementation. Employing model-derived lifetimes and $NO_2$ to $NO_x$ conversion factors reduced the NMB for both hotspots and background regions.

The long lifetimes at the hotspots and within the pollution plumes imply that the sink term is small. Therefore, estimated emissions at the source are mainly dependent on the divergence and can be computed accurately regardless of the uncertainty of the lifetime estimations, as long as the numerical smearing effect can be compensated. The opposite however applies when deriving emissions in the background regions where the sink term is significant owing to the shorter lifetimes. Indeed, for deriving total emissions at the simulation domain and at country level, in the cases where there is no significant inflow or outflow of $NO_x$ through the boundaries, the emissions are directly proportional to the sink term and depend crucially on the lifetime estimate.

When handling real TROPOMI data, substituting the default TM5-MP a *a-priori* with a more refined version derived from advanced high-resolution CTM simulations will mitigate the underestimation of hotspot emissions. This study revealed that emissions estimated with the TROPOMI LOTOS-EUROS *a-priori* product were 34% higher over the hotspots compared to the default TM5-MP *a-priori*.

Lastly, the runs conducted to test the FDA applicability outside the usual TROPOMI overpass time exhibit extensive areas with negative emission artifacts for the 08:00 LT test, and a generalized emission overestimation for the 18:00 LT test. These errors stem from ignoring the accumulation or depletion of atmospheric $NO_x$ during non-stable conditions, thereby invalidating the steady-state assumption of the FDA, and by remaining biases in the lifetime estimates. These findings suggest that the application of the FDA method to observations from geostationary sensors, such as TEMPO, GEMS, and the forthcoming

Sentinel-4 sensor, is more complex than applications during noontime observations, such as those from TROPOMI. In such cases, it is necessary to disregard the steady-state assumption and include the $dV/dt$ term to account for the buildup or de-
pletion of atmospheric $NO_x$. Furthermore, losses due to deposition and chemical reactions beyond the formation of $HNO_3$, such as oxidation leading to the production of alkyl and multifunctional nitrates, may play a more significant role, hence the assumptions underlying the $NO_2$ lifetime estimates might not be valid.

*Code and data availability.* The code V1.0 used in this paper to perform the end-to-end test of the flux divergence approach is accessible on Zenodo (https://doi.org/10.5281/zenodo.13268654, Cifuentes et al., 2024), along with a sample of the LOTOS-EUROS modeling outputs
required to run the code. The complete set of modeling outputs can be provided upon request. The TROPOMI L2 NO2 datasets used in this study are publicly available through the Copernicus Data Space Ecosystem (https://dataspace.copernicus.eu, last accessed: 08 August 2024).

## Appendix A: Statistical performance metrics

$$NMB = \frac{\sum_{i=1}^{n} P_i - O_i}{\sum_{i=1}^{n} O_i} \tag{A1}$$

$$NMGE = \frac{\sum_{i=1}^{n} |P_i - O_i|}{\sum_{i=1}^{n} O_i} \tag{A2}$$

Where n represents the total number of pixels, P is the predicted emissions at a given pixel i, and O is the original emissions ingested in the CTMs for pixel i.

## Appendix B: Supporting figures

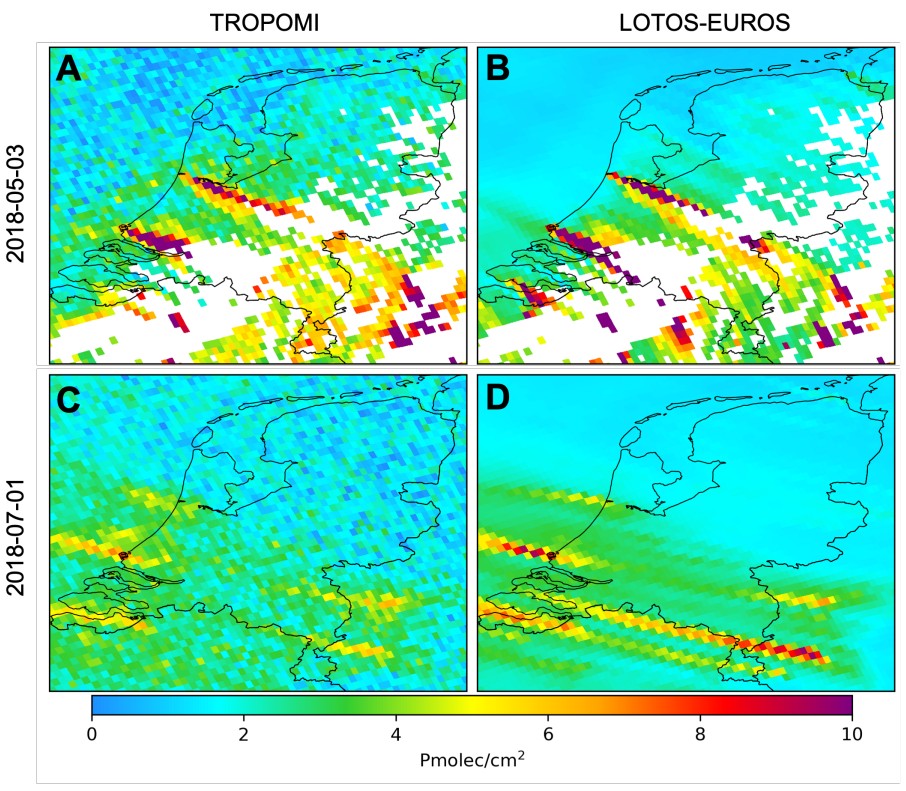

**Figure A1.** Daily $NO_2$ tropospheric vertical column densities over the Netherlands, as observed by TROPOMI and simulated using LOTOS-EUROS. **(A)** TROPOMI $NO_2$ VCD for May 3, 2018. **(B)** LOTOS-EUROS $NO_2$ VCD for May 3, 2018. **(C)** TROPOMI $NO_2$ VCD for July 1, 2018. **(D)** LOTOS-EUROS $NO_2$ VCD for July 1, 2018.

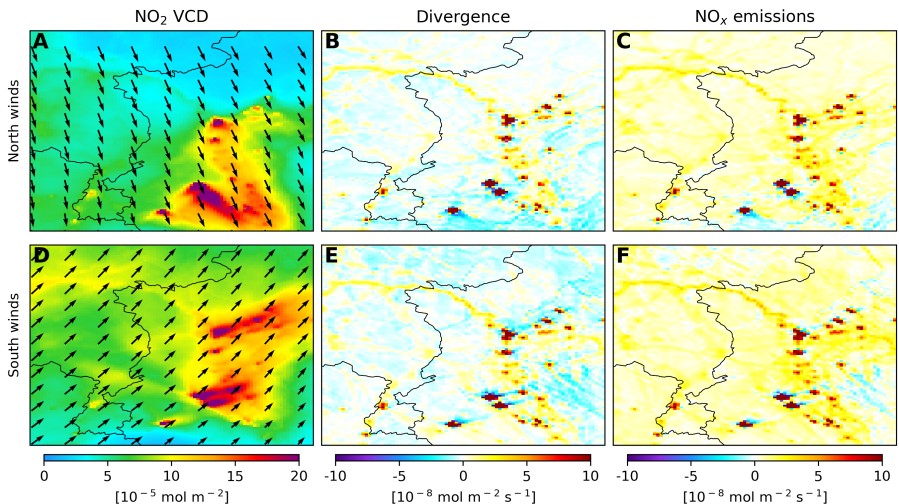

**Figure A2.** Mean $NO_x$ emissions and flux divergence derived from the synthetic dataset at LOTOS-EUROS native resolution using configuration ID07, aggregated by predominantly southerly and northerly winds. **(A)** Mean $NO_2$ vertical column density for days with predominantly north winds. **(B)** Mean $NO_x$ flux divergence for days with predominantly north winds. **(C)** Mean $NO_x$ emissions for days with predominantly north winds. **(D)** Mean $NO_2$ vertical column density for days with predominantly south winds. **(E)** Mean $NO_x$ flux divergence for days with predominantly south winds. **(F)** Mean $NO_x$ emissions for days with predominantly south winds.

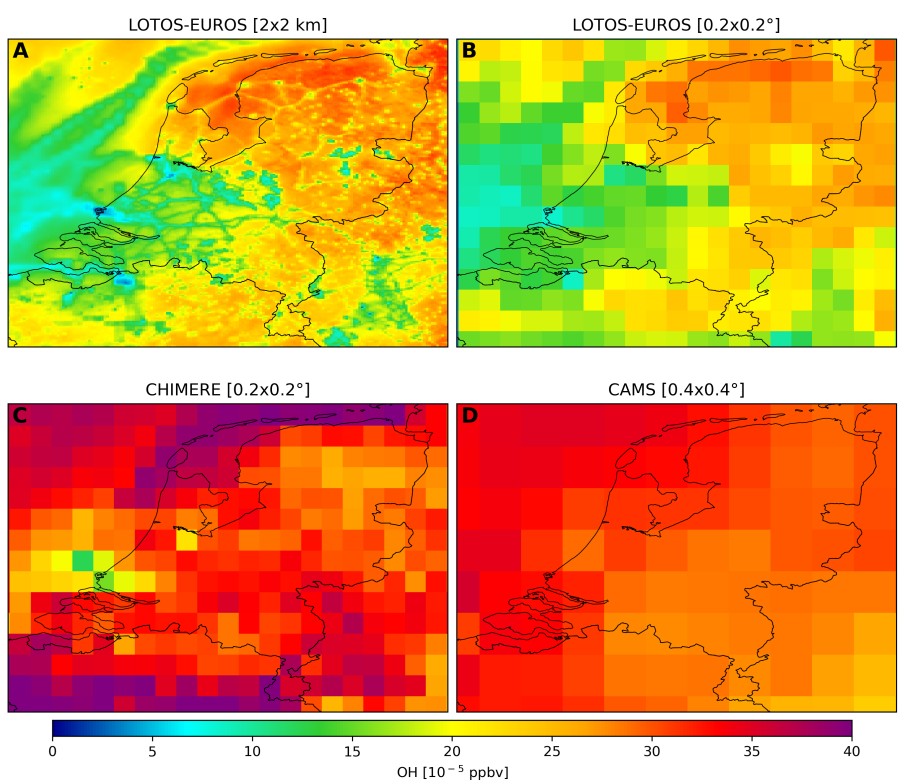

**Figure A3.** Mean ground OH concentration for JJA 2018 at 12:00 LT. **(A)** LOTOS-EUROS run at 2x2 km. **(B)** LOTOS-EUROS run at 0.2x0.2°. **(C)** Chimere run at 0.2x0.2°. **(D)** CAMS forecast at 0.4x0.4°.

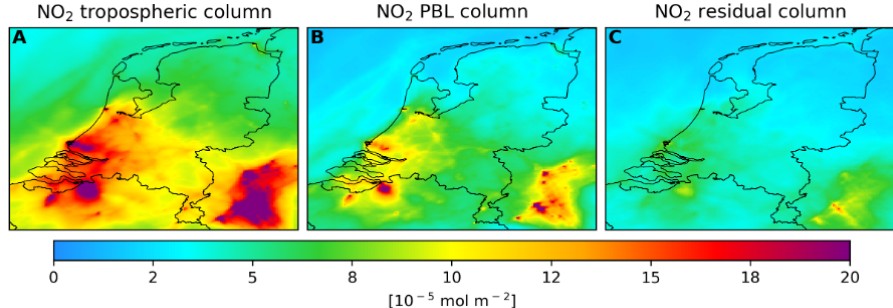

**Figure A4.** Mean $NO_2$ columns integrated up to the tropopause **(A)** and within the PBL **(B)**, and residual column **(c)** over the Netherlands for June, July, and August at the typical TROPOMI overpass time.

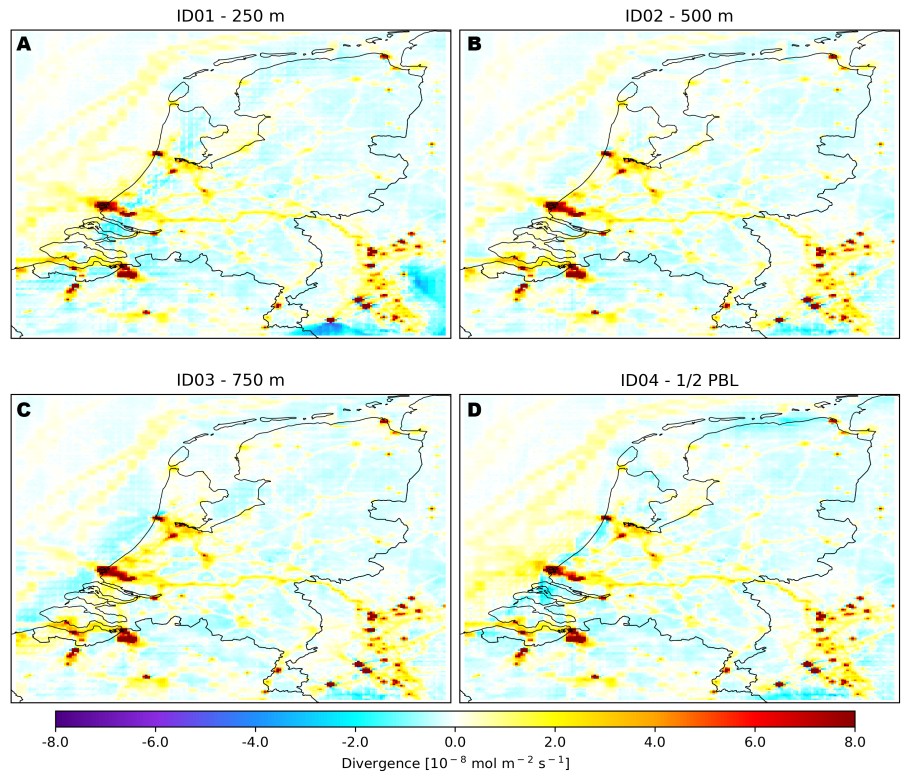

**Figure A5.** Mean $NO_x$ flux divergence for June, July, and August (JJA) using different altitudes for the wind fields. **(A)** 250 m. **(B)** 500 m. **(C)** 750 m. **(D)** Winds extracted at half planetary boundary layer height altitude

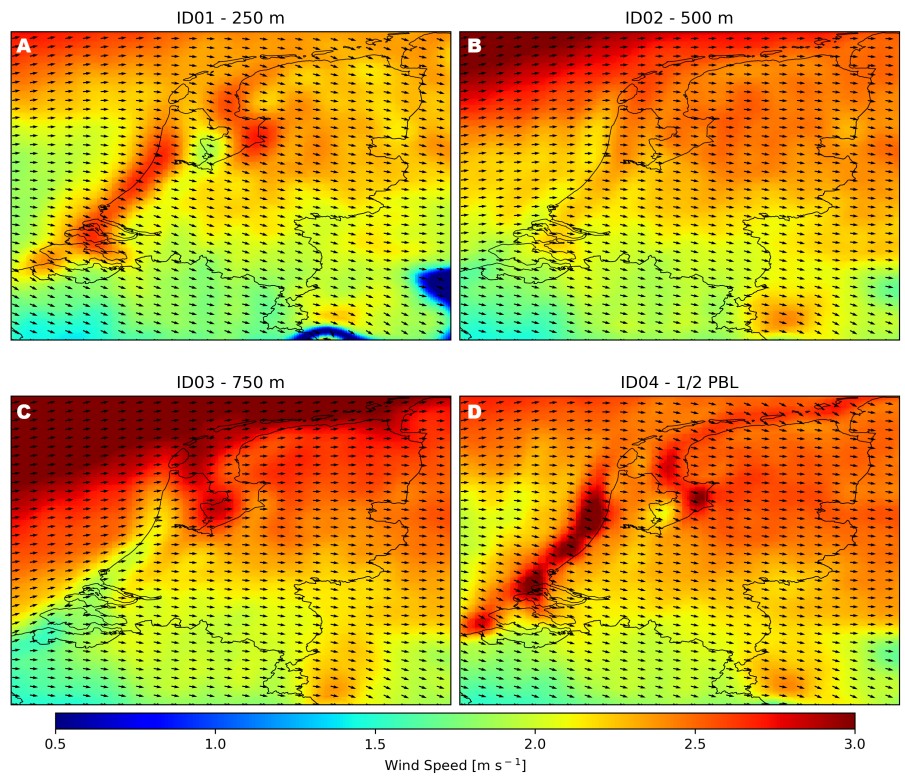

**Figure A6.** Mean wind fields for June, July, and August (JJA) at different altitudes. **(A)** 250 m. **(A)** 500 m. **(A)** 750 m. **(A)** Winds extracted at half planetary boundary layer height altitude.

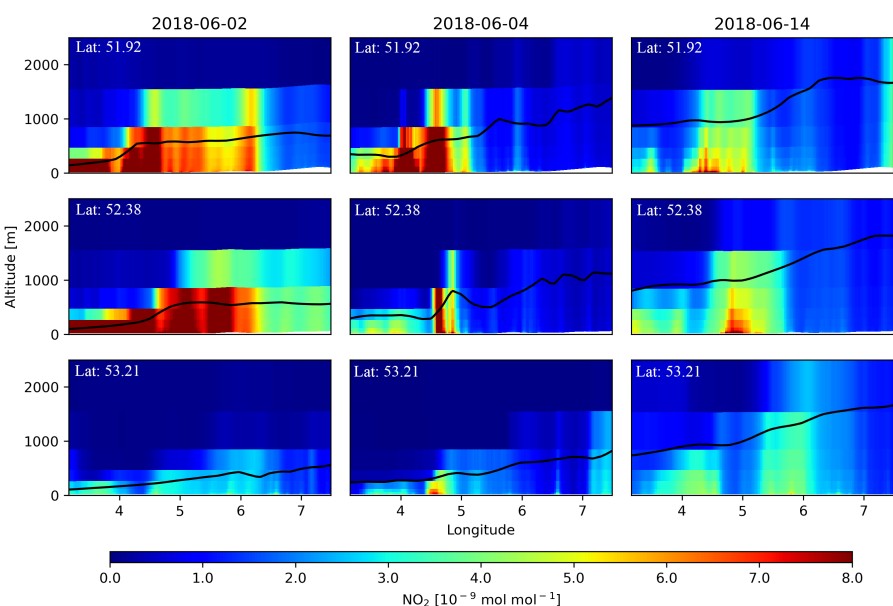

**Figure A7.** Latitudinal cross-sections of planetary boundary layer height and NO₂ concentrations at 13:30 LT extracted from LOTOS-EUROS simulation.

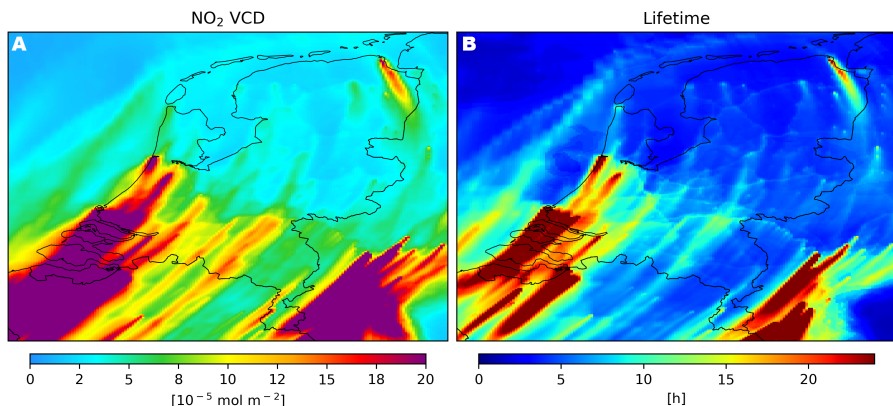

**Figure A8.** Example of $NO_2$ lifetime on a daily basis at satellite overpass time. **(A)** $NO_2$ vertical column density for a given day at satellite overpass time. **(B)** $NO_2$-weigthed lifetime derived from LOTOS-EUROS OH and T.

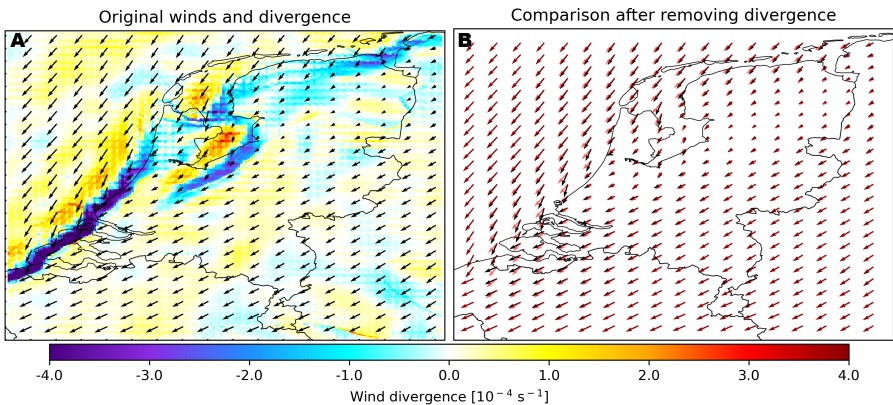

**Figure A9.** Example of ECMWF-IFS wind fields before and after reducing the divergence of the wind. **(A)** Original wind fields and corresponding wind divergence for a given day. **(B)** Original wind fields (black) and divergence-free wind fields obtained using the iterative algorithm proposed by Bryan (2022) (red).

*Author contributions.* **Felipe Cifuentes:** Methodology, Software, Formal analysis, Writing - Original Draft, Visualization. **Henk Eskes:** Conceptualization, Validation, Writing - Review and Editing, Supervision, Project administration, Funding acquisition. **Folkert Boersma:**
Conceptualization, Validation, Writing - Review and Editing, Supervision. **Enrico Dammers:** Methodology, Writing - Review and Editing. **Charlotte Bryan:** Conceptualization, Software.

*Competing interests.* The authors declare that they have no known competing financial or personal interests that could have influenced the work reported in this paper.

*Acknowledgements.* This study was funded by the Dutch Ministry of Agriculture, Nature and Food Quality (LNV), within the framework of
the National Nitrogen Knowledge Programme (NKS), project NKS-SAGEN on satellite observations and ensemble modeling.

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
