# Peer review of "Accurate space-based NOx emission estimates with the flux divergence approach require fine-scale model information on local oxidation chemistry and profile shapes"

_EGUsphere, 2024_

## Author Comment (AC2)

November 22, 2024

**Response to reviewers**
Ms. No.: egusphere-2024-2225
Geoscientific Model Development (GMD)

Please find attached our response to the reviewers' comments on the manuscript titled "Accurate space-based NOx emission estimates with the flux divergence approach require fine-scale model information on local oxidation chemistry and profile shapes" by F. Cifuentes, H. Eskes, F. Boersma, E. Dammers, and C. Bryan.

We sincerely thank the reviewers and the editor for their time and valuable feedback, which has significantly contributed to the improvements in this version of the manuscript. We have carefully reviewed all comments and have addressed them as outlined below.

Sincerely,

On behalf of all co-authors
**Felipe Cifuentes**
PhD Student
Royal Netherlands Meteorological Institute (KNMI)
Wageningen University & Research (WUR)
Email: felipe.cifuentescastano@knmi.nl
Telephone: +31611298892

**Reviewer Gerrit Kuhlmann**

The authors present a comprehensive analysis of the flux divergence method (FDA) using model fields created with the LOTOS-EUROS model for the Netherlands. The paper provides new insights in the accuracy of the FDA model. The paper is written well and the structure is clear. The methods are outlined well and clear with a few open questions.

**Comment from the authors:**
We sincerely thank the reviewer for their thoughtful and positive evaluation of our work. Responses to the comments are provided below.

**General comments**

1. The conclusions on the accuracy of the FDA model depend on the LOTOS-EUROS simulations. The authors should provide additional information on accuracy of the simulations, LOTOS-EUROS settings (e.g. spatial smoothing) and the impact on the conclusions.

**Response:**
We have added references to studies that evaluate the performance of LOTOS-EUROS for forecasting tropospheric NO$_2$ columns and its overall performance. This information was included starting from Line 130 as follows:

*"Within CAMS, LOTOS-EUROS undergoes routine validation with in-situ observations and TROPOMI satellite data, and is evaluated against the other ensemble members (Peuch et al., 2022). Additionally, independent studies have demonstrated good agreement between the simulated tropospheric NO$_2$ columns and those measured by TROPOMI and ground-based remote sensing instruments in the Netherlands (Vlemmix et al., 2015) and over Greece (Skoulidou et al., 2021a). These studies found discrepancies ranging from 1% to 35% when comparing LOTOS-EUROS columns with measurements from MAX-DOAS instruments. LOTOS-EUROS has also participated in numerous model intercomparison studies showing overall strong performance (Bessagnet et al., 2016; Colette et al., 2017; Vivanco et al., 2018)."*

We have also added a new paragraph beginning on Line 130 and included Figure A1 in the supplementary material to present our comparisons between LOTOS-EUROS NO$_2$ VCD and TROPOMI observations:

*"We compared LOTOS-EUROS synthetic NO$_2$ VCD with TROPOMI daily observations, finding good agreement in the shape, direction, and extent of plumes for major hotspots in the Netherlands, North Belgium, and West Germany. Figure A1 illustrates two examples of these comparisons. The high-resolution simulations of our simulations oversample TROPOMI's resolution by a factor of two, providing a more detailed representation of the chemistry within the plumes."*

[Figure]

**Figure A1.** Daily NO$_2$ tropospheric vertical column densities over the Netherlands, as observed by TROPOMI and simulated using LOTOS-EUROS.

Lastly, we provided further details on the vertical distribution of emissions (see our response to Comment 7) and on the possible impact of numerical diffusion (smoothing) in NO$_2$ simulations (see our response to Comment 14).

**2.** It is unclear how emissions are released in the model. If NOx emissions are released at the surface, it is likely that NOx will remain within the PBL and thus the approach of considering only the PBL column is feasible. However, as real NOx emissions often occur at stacks, vertical emission profiles should be used, which may result in emissions occurring in the free troposphere. In this case, using only the PBL column may not be a valid approach, especially in the case study for the morning overpass times where the PBL height is low.

**Response:**

We have added the following statement starting in Line 152 to clarify that emissions were ingested using vertical profiles:

*"Emissions are also distributed vertically, with specific heights assigned on a sector-by-sector basis. In particular, industrial sources and public power stations have vertical distributions based on typical average stack heights. Further details can be found in Manders et al. (2021)."*

We have also added the following statement, beginning on Line 276, to explain why we consider the PBL-integrated columns a valid approach:

*"Under typical noon conditions, which correspond to the TROPOMI overpass time, emissions are generally contained within the PBL. Therefore, we consider using PBL-integrated columns to be a valid approach. However, in the early morning, when the PBL is still forming, some emissions may extend beyond the PBL, a phenomenon that is prone to occurring only during winter months."*

**3.** The section on the NOx partitioning factor (Section 3.1.5) is quite short and ignores recent studies that have used values different from 1.32 (see specific comments).

**Response:**

To include references to the recent studies section 3.1.5 has been expanded. The updated text is provided in our response to comment 12 below.

**Specific comments**

**4.** L65ff: The differentiation between "plume dispersion models" and "mass conservation" is oversimplified here. The assumption of mass conservation is also used for methods that are applied to single sources (e.g. cross-sectional flux, Gaussian plume inversion and integrated mass enhancement). The statement "no need to run a computationally expensive CTM" contradicts the main conclusion of the study that CTM simulations are needed (last sentence in the abstract).

**Response:**

We have restructured the section from line 65 onward to expand on the distinctions between plume dispersion and mass conservation methods. Additionally, we acknowledge that the phrase "no need to run a computationally expensive CTM" may cause confusion and have removed it from the text. However, in the context of this study's results, the CTM serves as a forward model to provide information about $NO_2$ lifetime and $NO_X/NO_2$ partition ratios, which is still computationally less expensive than running a CTM inversion.

The previous text was:

*"CTM-independent approaches are based on plume dispersion models or mass conservation applying a steady-state continuity equation. Mass conservation approaches have several advantages, as there is no need to run a computationally expensive CTM, and in contrast to the plume dispersion approach, they enable the estimation of emissions at the satellite pixel resolution, providing information from multiple sources at once (Misra et al., 2021)."*

We have revised the text as follows:

*"CTM-independent approaches rely on either plume dispersion models or local mass conservation methods, applying a steady-state continuity equation. Plume dispersion methods involve fitting a dispersion model to concentration data across the impacted area to estimate emissions, which requires a precise definition of the region influenced by plume advection. In contrast, local mass conservation methods operate at the source or pixel level, eliminating the need to define a specific region of interest and enabling simultaneous estimation of multiple sources (Misra et al., 2021)."*

**5.** L75f: There have been some studies that have analyzed the FDA accuracy using model data: e.g. Goldberg at. 2022 and Hakkarainen et al. 2022.

**Response:**

We acknowledge the studies mentioned and have added the following sentence beginning on line 76 of the manuscript:

*"In addition, few studies have evaluated the accuracy and limitations of FDA satellite-derived emissions by using synthetic observations generated through chemical transport models as input within their approach (Dix et al., 2022; Hakkarainen et al., 2022; Goldberg et al., 2022)."*

**6.** L109f: You could mention here, why updating the a priori profile only partially corrects the bias.

**Response:**
We have added the following statement in line 113:

*"Remaining biases in TROPOMI observations might arise from errors in the slant column, the stratospheric-tropospheric partitioning, and various factors affecting the AMF beyond the a priori profile, including surface albedo and cloud cover."*

**7.** L137: Please clarify if emissions were released at the surface or vertical profiles were used. If emissions were released at the surface, it is likely that all NOx remains in the PBL. However, vertical profiles can release NOx into the free troposphere (in particular for low PBL), which has implications on the performance of using PBL columns only.

**Response:**
We have added the following statement starting in Line 152 to clarify that emissions were ingested using vertical profiles:

*"Emissions are also distributed vertically, with specific heights assigned on a sector-by-sector basis. In particular, industrial sources and public power stations have vertical distributions based on typical average stack heights. Further details can be found in Manders et al. (2021)."*

**8.** L161ff: Koene et al. 2024 show that divergence should be computed over the smallest region possible to avoid noise negatively affecting the divergence calculation (recommendation 5). A forth-order difference is therefore likely less ideal. You probably do not see the impact here, as you do not include noise in your $NO_2$ fields. However, I think it would be good to note that for application to noisy satellite images, a lower-order operator might be better.

**Response:**
Starting in line 187, we added the following statement to acknowledge the recommendation by Koene et al. (2024):

*"Koene et al. (2024) recommend calculating the divergence term using the smallest possible stencil to reduce noise impact. Consequently, applying a second-order finite difference method or estimating flux at cell boundaries might be more effective when processing noisy data. However, these improvements were not discernible in this study, as the synthetic $NO_2$ fields used are noise-free."*

**9.** L173ff: While NOx concentrations are stable around TROPOMI overpass, increasing turbulent mixing can still badly break the steady-state assumption inside plumes resulting in biased divergence fields.

**Response:**

We have complemented the statement starting in line 188 to incorporate this recommendation:

*"The steady-state assumptions imply the absence of accumulation or depletion of atmospheric NOx concentrations within the analyzed area. Factors such as turbulent mixing, changes in wind patterns, variations in emission sources, and sinks can disturb this balance (Koene et al., 2024). However, when averaged over time and across different realizations of the turbulence, the influence of these variations can be reduced".*

**10.** L180ff: Since the $NO_2$ enhancement will always have a vertical extent, the effective wind speed should be computed using the concentration profile of the $NO_2$ enhancement. Half the PBL height is a good approximation of the mean wind speed inside the PBL assuming well-mixed $NO_2$ concentrations, which isn't a bad assumption for cities or a few kilometers downstream of stack source (e.g. Krol et al. 2024).

**Response:**

We have restructured the paragraph starting in line 203 to include the limitation of the half PBL strategy as pointed out by the reviewer:

The previous text was:

*"The application of the FDA requires the reduction of the three-dimensional transport of pollutants in the atmosphere into a two-dimensional space. Therefore, zonal and meridional wind fields (or along and across track when using satellite grids directly) must be extracted at an altitude representative of the $NO_2$ bulk height. To achieve this, some authors have proposed selecting a fixed altitude lower than the usual PBL height during the satellite overpass time, such as 450 m (Beirle et al., 2019), 300 m (Beirle et al., 2021), 100 m (de Foy et al., 2022; Goldberg et al., 2022) and 80 m (Misra et al., 2021). Nonetheless, this approach neglects the day-to-day variability of the PBL and its spatial pattern. Bryan (2022) proposed to dynamically select the altitudes based on a fraction of the PBL height itself (half PBL height), while Lorente et al. (2019) used $NO_2$ weighted and unweighted mean boundary layer wind fields. These former methodologies try to capture the spatiotemporal variability of the PBL, and thus the effective height of the pollution plumes."*

We have revised the text as follows:

*"The application of the FDA requires the reduction of the three-dimensional transport of pollutants in the atmosphere into a two-dimensional space, involving the estimation of effective zonal and meridional wind fields (or along and across track when using satellite grids directly). These effective wind components should be column-weighted, which requires previous knowledge of the $NO_2$ and wind profiles within the column (Koene et al., 2024). As an approximation, Lorente et al. (2019) used $NO_2$ weighted and unweighted mean boundary layer wind fields, while Bryan (2022) proposed to dynamically extract the wind components based on a specific fraction of the PBL (half PBL height), which works well under the assumption of well-mixed $NO_2$ within the PBL. Alternatively, some authors proposed using a fixed altitude below the PBL height at the time of the satellite overpass, such as 450 m (Beirle et al., 2019), 300 m (Beirle et al., 2021), 100 m (de Foy et al., 2022; Goldberg et al., 2022) and 80 m (Misra et al., 2021). Nonetheless, this approach neglects the day-to-day variability and spatial patterns of the PBL, affecting the $NO_2$ and winds vertical distribution within the columns."*

**11.** L187f: The wind divergence will remain zero for a total column (assuming incompressible air), but not for partial column (e.g., PBL column), because air can leave or enter at the top of the partial column. In theory, it is possible to compute a two-dimensional wind, but this would require that you know both the $NO_2$ and wind profile (Koene et al. 2024). Thus, errors in the wind are caused by using (a) only a partial $NO_2$ column and (b) the wind field at a single (spatial varying) altitude.

**Response:**

We agree. In our previous response, we referred already to Koene et al. (2024) related to the effective wind aspect. One of our concerns was the impact of wind interpolation onto the grid and the choice of the wind field at a fixed altitude. Both factors contribute to numerical noise and fine-scale air mass divergences. This motivated us to study the wind divergence term separately from the column divergence, allowing us to better assess its influence on the quality of the estimated emissions.

**12.** L222f: The paragraph should acknowledge recent studies that have used different NOx partitioning factors: "Often a constant NOx:$NO_2$ ratio is assumed to infer the NO emissions from space-based NO observations. Many studies (e.g., Beirle et al., 2011, Beirle et al., 2019, de Foy and Schauer, 2022, Merlaud et al., 2020, Shaiganfar et al., 2017, Ionov et al., 2022, Potts et al., 2022, Hakkarainen et al., 2021) use the steady-state noontime molar concentration ratio under typical urban conditions of 1.32 based on Seinfeld and Pandis (2006). Recently, model-based concentration ratios have also been calculated using simulations from Copernicus Atmospheric Monitoring Service (CAMS, Lorente et al., 2019, Rey-Pommier et al., 2022) and Comprehensive Air Quality Model with Extensions (CAMx, Goldberg et al., 2022). Beirle et al. (2021) calculated NOx:$NO_2$ ratios according to the photo-stationary steady state. In general, these studies show small deviations from the value 1.32 (e.g., 1.16–1.83), but acknowledge that values near the point sources are likely to be higher. CHIMERE model simulations (Shaiganfar et al., 2017) further indicate that in large circles around Paris, the partitioning ratios are smaller during summer (1.32) than in winter (1.51), due to the higher ozone mixing ratios in summer. In contrast to model-based analyses, the Dutch aircraft measurements of in-plume NOx/$NO_2$ ratios from power stations (e.g., Janssen, 1988, Vilà-Guerau de Arellano et al., 1990, Bange et al., 1991, Hanrahan, 1999) often showed values higher than 10 near the source and values between 2 and 10 up to 15 km from the source." (Hakkarainen et al. 2024).

**Response:**

The paragraph starting in line 251 has been expanded as follows:

*"Similarly to τ, the partition between NO and $NO_2$ in the atmosphere is influenced by factors such as the actinic flux, ozone concentrations, and temperature. A constant mean value of L = 1.32 is often assumed (Beirle et al., 2019; de Foy et al., 2022; Misra et al., 2021), which is considered representative of the usual satellite observation conditions (noon time and cloud-free pixels). However, recent studies increasingly use model-derived values for L. These include simulations from global models like CAMS (Lorente et al., 2019; Rey-Pommier et al., 2022) and regional models (Goldberg et al., 2022). Alternatively, Beirle et al. (2021) derived L values using a photo-stationary steady-state approach. These methods have reported L values ranging from 1.16 to 1.83, deviating from the typical mean of 1.32. Large eddy simulations (LES) further indicate that L can rise as high as 5 within the first 10 km of emitted plumes (Krol et al., 2024)."*

**13.** L245: I am also not aware of any studies that subtracted the background from $NO_2$ observations. The background has also been subtracted from CO2 columns by Hakkarainen et al. (2022). Koene et al. (2024) show that removing the background eliminates the steady-state assumption for the background component.

**Response:**

We did a small addition in the statement starting in Line 279 to also cite Hakkarainen et al. (2022) for subtracting the background from the columns:

*"To the best of our knowledge, applications of the FDA approach in the peer-reviewed literature have exclusively used $NO_2$ tropospheric columns without critically considering that free tropospheric $NO_2$ is decoupled from surface emissions. Liu et al. (2021) excluded the free tropospheric contribution in their analysis of methane emissions, while Hakkarainen et al. (2022) applied a similar approach for $CO_2$."*

Additionally, we incorporated the findings of Koene et al. (2024) into the discussion section, starting at line 389:

*"Furthermore, Koene et al. (2024) noted that removing background fields is advantageous because it allows the analysis of wind fields to be confined to the PBL rather than the entire column, and also eliminates the need to make assumptions about the steady-state conditions for the background field."*

**14.** L304ff: You earlier state that using a second-order difference instead of fourth-order difference had only a minor impact on your results. This contradicts your statement here that the fourth-order difference cause (strong) spatial smearing. Therefore, the smearing likely has a different explanation: I would expect some smoothing from LOTOS-EUROS depending on the model dispersion settings, as the effective model resolution is typically coarser than grid resolution of 2 km. It is also possible that LOTUS-EUROS is not mass conserving at strong point sources.

**Response:**

The smearing in the FDA can be caused by the numerical method used to resolve the divergence term, as it effectively incorporates information from neighboring grid cells to estimate changes at the grid level. Additionally, as the reviewer pointed out, the simulated fields from the CTM are also prone to numerical diffusion when resolving the advection equation on the model grid, which smooths the $NO_2$ plumes. We have added the following statement in Line 343 to highlight numerical diffusion within the CTM as a plausible cause for the smearing in our FDA results:

*"The smearing effect may also result from biases in the CTM-predicted $NO_2$ fields due to numerical diffusion when resolving the advection equation at the model grid, which artificially smooths the solution in regions with steep concentration gradients, leading to plume stretching (Eastham and Jacob, 2017; Rastigejev et al., 2010)."*

**15.** L335ff: Is using the PBL column still feasible, if strong point sources release in the free troposphere in particular for low PBLs? Was this include in the model simulations?

**Response:**

As previously mentioned in response to reviewer comment number 7, emissions were incorporated into the model using vertical profiles specific to each sector, taking into account

typical average stack heights for industrial and energy sector emissions. Given that the method is applied under noon conditions (TROPOMI overpass time), emissions are expected to be confined within the PBL, apart from a few exceptions in winter with extremely stable conditions, making the use of a PBL-integrated column appropriate.

For other sensors, such as GEMS and TEMPO, which provide high-resolution measurements during the morning hours, some emissions may be released above the planetary boundary layer (PBL), which remains shallow and under development during this period.

**16.** L384f: As noted above, many recent studies used values other than 1.32.
**Response:**
Agreed, references to several studies have been incorporated into the manuscript, as detailed in the response to comment 12.

**17.** L401f: The impact of the divergence-free winds is likely small, because you subtract the background from the $NO_2$ column. Koene et al (2024) show that omitting the wind divergence term is useful if the background is not removed from the column.
**Response:**
We have added a statement starting on Line 456 to contextualize the findings of Koene et al. (2024):

*"The small effect of reducing the wind divergence term in our results is due to our integration of $NO_2$ VCD only up to the PBL height. As noted by Koene et al. (2024), addressing the wind divergence term can be beneficial when the background is not subtracted from the columns."*

**18.** L416f/L440ff: Do you use only PBL columns in this case study? I would expect that in the morning the PBL is very low, which would "leak" some $NO_2$ in the free troposphere and thus result in an underestimation of the emissions.
**Response:**
The PBL-integrated $NO_2$ column was also used to evaluate the FDA at 08:00 and 18:00 local time. In the early morning case, PBL heights are indeed lower, allowing emissions from particularly high stacks, such as those in the energy sector, to potentially exceed the PBL. However, as shown in Figure 8B of the manuscript, emission hotspots are accurately represented, and negative artifacts appear in "clean" areas, away from stack emissions. Therefore, the observed underestimation of emissions is not due to $NO_2$ extending above the PBL but rather, as noted in the manuscript, due to violations of the steady-state assumption at this time and biases in the estimated $NO_2$ lifetime.

**19.** L486: An underestimation of the emissions by 18% is still larger than 11% from the synthetic data. Can you provide a brief a discussion for the reasons? Is the bias due to overestimated emissions in simulation or caused by the FDA method?
**Response:**
We have added the following statement starting on Line 555

*"Using the bottom-up inventory as the baseline, TROPOMI-derived emissions are 32,2% higher for the entire domain, potentially due to TROPOMI signal issues, non-negligible*

*accumulation terms (dV/dt), or differences in soil emissions. On the contrary, when considering only hospot areas, emissions are biased low by 18%. Applying the FDA to the synthetic data (see section 4.2) revealed an 11% negative bias at hotspot locations, indicating a general tendency of the FDA to underestimate emissions at these points. The additional 7% bias observed when using real TROPOMI data with the updated a-priori falls within the uncertainty ranges of both TROPOMI and bottom-up inventories."*

**20.** L530f: Most studies use ERA-5 wind fields instead high-resolution simulations. Do you expect an impact on estimated emissions using ERA-5 fields?
**Response:**
We included a paragraph starting on Line 610 discussing the potential impact of using a coarser-resolution wind dataset:

*"In this study, ECMWF-IFS wind fields, with an original resolution of roughly 9 km, were downscaled to a high spatial resolution of about 2 x 2 km. Nonetheless, the FDA is often applied using coarser publicly available datasets, such as ERA5, with a horizontal resolution of 0.25 x 0.25 degrees. Such coarse datasets may not accurately capture the spatial variability of wind fields, especially in regions with complex topography, which can introduce artifacts in divergence estimation. In these scenarios, using higher-resolution wind fields is recommended."*

**21.** L553: What are the implications for Sentinel-5 with an overpass time of 9:30 LT?
**Response:**
The following statement was added starting in line 638:

*"In such cases, it is necessary to disregard the steady-state assumption and include the dV/dt term to account for the buildup or depletion of atmospheric $NO_X$. Furthermore, losses due to deposition and chemical reactions beyond the formation of $HNO_3$, such as oxidation leading to the production of alkyl and multifunctional nitrates, may play a more significant role, hence the assumptions underlying the $NO_2$ lifetime estimates might not be valid."*

**Technical corrections**

**22.** L6: Summer -> summer
**Response:**
The correction has been made in Line 22

**23.** L98: DOAS -> Differential Optical Absorption Spectroscopy (DOAS)
**Response:**
The correction has been made in Line 98

**24.** Figure 3: Mention that the figure is for ID06.
**Response:**
The caption of Figure 3 has been modified to "Comparison between the original and convoluted $NO_X$ model-ingested emissions and FDA-derived emissions for June, July, and August (JJA), at 13:30 LT, using configuration ID06."

**25.** Table 3: Please add units.
**Response:**
Thank you for your suggestion. The metrics in Table 3 (Normalized Mean Bias and Normalized Mean Gross Error) are dimensionless, as they represent normalized values. Therefore, no units are applicable.

**Reviewer Anonymous Referee #2**

This paper discusses NOx emissions using the flux-divergence approach (FDA) on measurements from space-based instruments, primarily TROPOMI. Specifically, the sensitivity of the derived emissions to the $NO_2$ lifetime, $NO_X$-$NO_2$ ratio, column integration height, wind altitude, and wind divergence were tested. The researchers used the LOTOS-EUROS chemistry transport model at 2x2 km to produce high-resolution values for these various parameters, then compared to fixed values. For example, the varying $NO_2$ lifetimes produced from LOTOS-EUROS were used in one test while a fixed 4-hour lifetime was used in another.

The study offers valuable contributions for understanding the FDA and its limitations in this application. The various types of comparisons—first with synthetic observations on the model grid, then on the TROPOMI on grid, and finally with TROPOMI observations—helped distinguish which factors affected the error between the model-ingested emissions and the derived emissions.

**General Comments**

1. More discussion is advised for the section on real TROPOMI data, where the a priori TM5 profiles are replaced with profiles from LOTOS-EUROS. Since most research will focus on real observations impacted by the retrieval process, the implications of this a priori replacement were underdiscussed. The paper often lacks citation to other related work (especially research on the role of high space and time resolution a priori profiles) which would provide context to the researchers' experiment design and findings.

**Response:**
We have included additional information on the impacts of a-priori replacement in TROPOMI retrievals, both in a general context and specifically within the scope of our study. Please refer to our response to Comment 15 for further details. In addition, in response to Referee 1, we have expanded the references to existing studies addressing aspects such as wind fields, chemical lifetimes, and the $NO_2$/NOx ratio.

**Specific Comments**

2. Line 78: As discussed in Douros et al. (2023), a major contribution to this bias is also the a priori $NO_2$ profile used in the retrieval. This is related to, but different from, the issue of reduced sensitivity near the surface.

**Response:**
We agree and have included the a priori $NO_2$ profiles as a potential source of bias in $NO_2$ retrieval. Changes were made starting on Line 80:

*"The major sources of uncertainty in the method are considered to be biases in the observed $NO_2$ VCD, due to reduced vertical sensitivity to near-surface $NO_2$ by the satellite products, and the influence of selected a-priori $NO_2$ vertical profile shapes (Douros et al., 2023)."*

Further discussion on the impact of replacing TROPOMI a priori is provided in Section 4.3.

**3.** Line 130: It would be helpful to clarify how the top vertical layer compares to the typical location of the tropopause, since the synthetic observations are compared with tropospheric columns.

**Response:**
We have added the following statement starting on line 139 to clarify this point:

*"The simulations were conducted using 12 vertical levels, extending from the ground to approximately 9 km above the earth's surface. Tropopause heights can exceed 15 km in tropical regions and typically range between 8 and 12 km in other areas (Xian and Homeyer, 2019). Given that this study focuses on the Netherlands, using vertical layers up to 9 km should provide a reasonable estimate of the tropospheric $NO_2$ column, as the majority of tropospheric $NO_2$ is contained within this altitude range."*

**4.** Line 210: Running a CTM at fine resolution has also been shown to reduce $NO_2$ lifetimes in NOx-limited regimes (e.g. Li et al., 2023).

**Response:**
We expanded the statement starting on Line 236 to include the suggested reference:

*"A CTM running at fine resolution can result in extended lifetimes in NOx-saturated regions due to enhanced OH titration by NOx. Conversely, in NOx-limited regimes, it can result in shorter lifetimes as elevated VOC levels promote OH production in the presence of the available NOx (Li et al., 2023; Krol et al., 2024)"*

**5.** Line 215: Notably, the study domain of Rey-Pommier et al. (2022), is Egypt, which has comparatively little forestation. Delaria et al. (2020), indicates that the lifetime to stomatal deposition can be as short as 10 hours in forested regions. The expected lifetime to deposition in the study domain (Netherlands) should be explored.

**Response:**
In our simulations using the LOTOS-EUROS model, we can estimate $NO_2$ dry deposition fluxes, as the model incorporates the Deposition of Acidifying Compounds (DEPAC) 3.11 module, which uses a resistance-based approach (van Zanten et al., 2010).

Figure 1 in the response to reviewers shows the average NOx emission flux and $NO_2$ dry deposition flux at the typical satellite overpass time for the period analyzed (June, July, and August), as derived from the LOTOS-EUROS simulation. The dry deposition fluxes are an order of magnitude lower than the emission fluxes. In fact, across the simulation domain, the dry deposition flux is approximately 5% of the total $NO_X$ emissions flux. This contribution decreases further when focusing on hotspot areas. Therefore, including an associated lifetime for deposition would likely not significantly affect our emission estimates at the typical TROPOMI overpass time. While the relative contribution of deposition versus total emissions may increase at other times of the day, this falls outside the scope of our study.

[Figure]

**Figure 1.** Mean NOx emissions and NO₂ dry deposition fluxes over the Netherlands for June, July, and August at the typical TROPOMI overpass time.

**6.** Line 255: The definition of a "hotspot" is set at 10% percentile of the original emission inventory. This definition is later used to draw key findings: for example, when comparing the use of TM5 vs. LOTOS a priori profiles, TM5 has a smaller positive bias overall but much greater negative bias among hotspots (Table 3). To strengthen conclusions based on the "hotspot" division, the choice of 10% could be further discussed. How sensitive are the study's conclusions to the choice of percentile? Is there another reason to emphasize the accuracy at these hotspots at the expense of the domain more broadly?

**Response:**

Hotspots were defined as the pixels corresponding to the top 10% of emissions in the original emission inventory, on a pixel-by-pixel basis. Although these hotspots represent only a small fraction of all pixels, they encompass over 56% of total emissions within the simulation domain, thus capturing the primary emission sources. Consequently, it is crucial to prioritize these areas in our analysis.

Additionally, it is essential to draw conclusions separately for hotspots and the entire domain, as background and hotspot regions respond differently to certain parameters within the FDA. For example, the divergence term reaches its maximum at the source locations, while contributions from lifetime or sink terms become small. In contrast, the background shows the opposite behavior. Focusing on the hotspots enables assessment of the method's performance in areas where the majority of emissions are concentrated.

To illustrate the sensitivity of our results to the chosen percentile for defining hotspots, we have included Table 1 in this response to the reviewer. This table presents performance metrics using the synthetic observations at LOTOS-EUROS grid using configuration ID06, with hotspots defined as the top 10%, 15%, and 20% of highest-emission pixels. As shown, the metrics exhibit minimal variation, particularly after emissions are convoluted, which serves as our reference point for drawing conclusions in this study.

**Table 1.** Performance metrics using the synthetic observations at LOTOS-EUROS grid using configuration ID06

| Hotspots percentile | Percentage of total emissions | Original emissions | | | Convoluted emissions | | |
|---|---|---|---|---|---|---|---|
| | | NMB | NMGE | R | NMB | NMGE | R |
| 10% | 56% | -28,3 | 43,7 | 0,92 | -8,6 | 22,3 | 0,96 |
| 15% | 64% | -24,7 | 42,9 | 0,91 | -8,5 | 23,3 | 0,96 |
| 20% | 71% | -21,4 | 43,2 | 0,88 | -7,8 | 24,2 | 0,96 |

**7.** Line 324: See note on Line 210
**Response:**
The statement in Line 362 was expanded as follows:

*"Higher resolution simulations exhibit an enhanced titration of OH with NOx in NOx-saturated regions, leading to an extended lifetime, while having the opposite effect in NOx-limited regions (Li et al., 2023; Krol et al., 2024)"*

**8.** Line 337: Were there any spatial trends in the proportion of free tropospheric $NO_2$?
**Response:**
We have included Figure A4 in the supplementary material of the manuscript and added the following paragraph, starting at line 380, to discuss the spatial patterns of free tropospheric $NO_2$:

*"The residual $NO_2$ column obtained after subtracting the PBL contribution (See Figure A4), still exhibits significant concentrations over the Ruhr area, Antwerp, and along the Dutch coastline. These regions contain the majority of NOx emissions, suggesting that the spatial patterns in the residual layer may reflect enhancements in the free troposphere caused by emissions from previous hours. The high concentration in the residual layer can also originate because the PBL does not align precisely with the model layer interfaces, leaving part of the column excluded from the inversion process. This can partially explain the low negative bias of -8% when deriving emissions using this approach."*

[Figure]

**Figure A4.** Mean $NO_2$ columns integrated up to the tropopause (A) and within the PBL (B), and residual column (C) over the Netherlands for June, July, and August at the typical TROPOMI overpass time.

**9.** Line 349: The researchers find limited sensitivity to the altitude selected for wind fields. Table 1 suggests that this was a key question, since 4 of the 7 tests focus on the variation of winds altitude. The discussion centers on how this may be unique to the terrain of the studied domain, but it is not clear if the researchers were limited to this domain or chose not to further explore the effect of a different terrain on this aspect of the FDA.

**Response:**

A primary source of uncertainty in applying the FDA approach arises from the wind fields used to derive the divergence term. To assess this impact, several tests were implemented. Our findings indicate that the choice of wind field strategy has limited influence, likely due to the relatively simple wind patterns over the flat terrain of the Netherlands. However, we recommend further investigation when applying the FDA approach in regions with more complex topography. Expanding this analysis beyond the Netherlands was not within the scope of our study for several reasons.

Firstly, high spatial resolution simulations are computationally intensive, restricting the analysis to smaller areas. Additionally, the simulations were carried out within the framework and with the funding of the Nitrogen Knowledge program (NKS; https://www.onslevendlandschap.nl/basis-herstellen/nks ), which focuses specifically on nitrogen deposition in the Netherlands, justifying the emphasis on this region. Finally, the Netherlands represents an ideal case study due to its elevated levels of NOx pollution; this analysis is therefore especially beneficial for understanding the temporal and spatial distribution of such emissions in this region.

**10.** Line 382: The need to accurately represent lifetime is not unique to the FDA method, especially for larger scale emission retrievals like those discussed above. Plume-based methods (e.g. in Beirle et al. (2011), Valin et al. (2013) and many, many other more recent references from an array of research groups) rely on simultaneous retrieval of both the lifetime and emissions from satellite observations, with the implications for the estimated emissions explored in-depth. The researchers should show how their findings about the lifetime's importance builds on previous approaches to deriving NOx emissions.

**Response:**

The manuscript discusses the applications and limitations of model-derived lifetimes in Section 4.1. Additionally, we have included the following statement, starting in Line 432, to further elaborate on the use of simultaneously derived emissions and lifetimes:

*"The complexities involved in estimating lifetimes are not unique to the FDA. Model-derived lifetimes, based on OH availability, can be biased, as previously discussed in Section 4.1. Alternative methods for deriving lifetimes include the simultaneous estimation of emissions and lifetimes using downwind $NO_2$ patterns derived from the satellite observations directly. This approach reduces reliance on prior assumptions or model-based inputs and is applied in plume dispersion methods to estimate emissions from single (Beirle et al., 2011; Valin et al., 2013) or multiple sources (Fioletov et al., 2022; Dammers et al., 2024). However, this method is not always feasible, as plumes are not always isolated and cannot always be idealized as point sources Additionally, covariance between emissions and lifetimes introduces errors into lifetime estimations."*

**11.** Line 433: "Furthermore, the planetary boundary layer (PBL) reaches its peak during this time, aiding in the dispersion and dilution of pollutants." This is one two reasons provided for the decreased in vertical column densities (VCDs) from 11:00 to 16:00 local time. While vertical mixing will increase at the peak of the PBL, this would not impact the integrated vertical column densities. There could be other reasons that the increased dispersion/dilution at this time would lead to decreased VCDs, but this should be clarified.

**Response:**
We agree. While vertical mixing alone does not alter the NOx column, it facilitates the mixing of VOCs and NOx, thereby enhancing OH production. To address this, we have added clarification starting at line 490:

*"Furthermore, the PBL reaches its peak during this time, enhancing the mixing of NOx and VOCs, which further promotes OH formation". The other factor contributing to the observed decrease in VCDs has already been discussed in the manuscript: the increased photochemistry driven by higher incoming solar radiation during this period, which helps $O_3$ formation and subsequently contributes to OH production."*

**12.** Line 445: While this is true for a single instrument, work has been done to estimate changes in vertical column using multiple polar-orbiting satellites (Penn and Holloway, 2020)

**Response:**
We have expanded the statement beginning in Line 501 to incorporate this insight:

*"However, for polar orbit satellites like TROPOMI, evaluating this term is not viable for a single instrument, as observations are limited to just one per day. Nonetheless, it would be possible to estimate changes in $NO_2$ VCD using multiple polar-orbiting satellites with similar overpass times. This approach has been demonstrated by Penn and Holloway (2020) using the GOME-2 and OMI instruments, as well as by Boersma et al. (2009) using SCIAMACHY and OMI."*

**13.** Line 448: What reactions are expected to start competing at these times? Generally, how are the assumptions about loss to OH complicated by the growing importance of $RONO_2$ chemistry (Romer Present et al., 2020)

**Response:**
We have revised the statement starting on Line 508 to elaborate on the potential bias in the $NO_2$ lifetime:

*"This discrepancy might stem from uncertainties in the OH estimates utilized for deriving the lifetime, or it could indicate the need to incorporate additional competing reactions that consume OH and other sink pathways for NOx, such as oxidation to produce alkyl and multifunctional nitrates (Sobanski et al., 2017; Romer Present et al., 2020). In addition, as photochemical processes diminish during this time of day, the contribution of NOx loss due to deposition could become more significant."*

**14.** Line 466: What are the temperature-dependent emissions in the model and what direction does this direct the resulting bias? (E.g. are these diesel related as in Grange et al. 2019?)

**Response:**

We have added the following statement on Line 529 to clarify the temperature-dependent emissions included in the LOTOS-EUROS simulation:

*"In LOTOS-EUROS, these temperature-dependant emissions include NOx emissions from soils, biogenic VOC (also influenced by photosynthetically active radiation), and road transport emissions (Manders et al., 2021). Additionally, the increase in incoming solar radiation under cloud-free conditions enhances the photolysis of several compounds."*

**15.** Line 488: When discussing agreement between FDA-derived emissions and model-ingested emissions, there should be more exploration on the effects of using a priori profiles from LOTOS-EUROS. For example, the model wind fields will impact the retrieved VCDs and not just the divergence term. The importance of using a priori inputs with accurate meteorology are discussed in Laughner et al. (2016) and would impact the gradients that are analyzed through the divergence term.

**Response:**
We have added two paragraphs in section 4.3 of the manuscript to expand on the impacts of a priori replacement in TROPOMI:

Starting in Line 535: *"Previous studies have shown that replacing the TROPOMI coarse TM5-MP $NO_2$ a-priori vertical profile with a high-resolution offers a more accurate representation of meteorological and chemical fields which increase column concentrations near emission sources by more than 30% and create a steeper concentration gradient around these areas (Laughner et al., 2016; Griffin et al., 2019; Zhao et al., 2020; Judd et al., 2020), leading to better agreement with ground-based validation (Douros et al., 2023; Liu et al., 2020; Ialongo et al., 2020). Based on these findings, we conducted two tests using TROPOMI with the default TM5-MP5 $NO_2$ a-priori profile shape, and a modified version incorporating an updated LOTOS-EUROS $NO_2$ profile."*

Starting in Line 549: *"The updated a-priori profile not only enhances $NO_2$ vertical column densities (VCDs) at hotspot locations but also provides a more accurate representation of concentration gradients, which impacts the estimation of the divergence term in the FDA (Laughner et al., 2016). Additionally, because our study uses consistent wind data for both generating the new a-priori profiles and calculating the divergence, we expect more accurate divergence estimates."*

**16.** Line 511: It is claimed that good agreement exists between model-ingested emissions and FDA-derived emissions when LOTOS-EUROS is used for the a priori profiles. The authors claim that "top-down satellite and bottom-up inventory emissions are derived in a independent way, except for the a-priori replacement." The change in a priori profiles will certainly have an effect on the retrieved VCDs, and therefore the emissions from the FDA. It would be more appropriate to say that the FDA produces values in agreement with known emissions that informed the retrieval. However, if the true emissions were not the same as model-ingested emissions, it is not clear how much agreement would be observed. While this is not necessarily the focus of the study, further elaboration on this point would be advised. This could also build on existing studies investigating the relationship between model-ingested emissions, model resolution, and top-down emissions estimates (e.g. Goldberg et al., 2019). This is important for readers interested in the value of replacing the

a priori profiles, but for the purpose of obtaining new emissions estimates rather than verifying an existing one.

**Response:**

We have revised the statement starting in Line 586 as follows:

*"This represents a good level of agreement, considering that the top-down satellite and bottom-up inventory emissions are fully independent pieces of information (observed versus reported emissions), and that the comparisons are done in a consistent way through the a-priori replacement."*

Additionally, we have added the following statement starting in Line 589 to address the impact and limitations of updating the a-priori:

*"The high-resolution model, incorporating prior knowledge of emission locations and strengths, provides important constraints on the spatial and temporal variability of $NO_2$ needed for quantitative comparisons. When bottom-up emissions significantly deviate from the real emissions, such as in cases of missing sources, this can result in errors in satellite-derived emissions of up to approximately 30%. These discrepancies can be reduced through iterative refinement, using satellite-derived emissions to update the model employed for generating the a a-priori estimates."*

**Technical Corrections**

**17.** Line 97: correct "NO2" to "$NO_2$" for consistency with rest of paper
**Response:**
The correction has been made in Line 97

**18.** Line 245: correct "FDA approach" to "FDA"
**Response:**
The correction has been made, and checked in the entire manuscript for consistency.

**19.** Line 515: correct "a independent" to "an independent"
**Response:**
The correction has been made.